# A TORC1-histone axis regulates chromatin organisation and non-canonical induction of autophagy to ameliorate ageing

Yu-Xuan Lu[1], Jennifer C Regan[2†], Jacqueline Eßer[1], Lisa F Drews[1], Thomas Weinseis[1], Julia Stinn[1‡], Oliver Hahn[1§], Richard A Miller[3], Sebastian Grönke[1], Linda Partridge[1,2]*

[1]Max Planck Institute for Biology of Ageing, Cologne, Germany; [2]Institute of Healthy Ageing, Department of Genetics, Evolution and Environment, University College London, London, United Kingdom; [3]Department of Pathology, University of Michigan, Ann Arbor, United States

**Abstract** Age-related changes to histone levels are seen in many species. However, it is unclear whether changes to histone expression could be exploited to ameliorate the effects of ageing in multicellular organisms. Here we show that inhibition of mTORC1 by the lifespan-extending drug rapamycin increases expression of histones H3 and H4 post-transcriptionally through eIF3-mediated translation. Elevated expression of H3/H4 in intestinal enterocytes in *Drosophila* alters chromatin organisation, induces intestinal autophagy through transcriptional regulation, and prevents age-related decline in the intestine. Importantly, it also mediates rapamycin-induced longevity and intestinal health. Histones H3/H4 regulate expression of an autophagy cargo adaptor Bchs (WDFY3 in mammals), increased expression of which in enterocytes mediates increased H3/H4-dependent healthy longevity. In mice, rapamycin treatment increases expression of histone proteins and *Wdfy3* transcription, and alters chromatin organisation in the small intestine, suggesting that the mTORC1-histone axis is at least partially conserved in mammals and may offer new targets for anti-ageing interventions.

*For correspondence: Linda.Partridge@age.mpg.de

Present address: †Institute of Immunology and Infection Research, University of Edinburgh, Edinburgh, United Kingdom; ‡CECAD Research Center, Cologne, Germany; §Department of Neurology and Neurological Sciences, Stanford University School of Medicine, Stanford, United States

Competing interests: The authors declare that no competing interests exist.

## Introduction

Ageing leads to the functional decline of cells, tissues, and organs, and is the primary risk factor for the most common, fatal human diseases, including cancer, cardiovascular disease, and neurodegeneration (*Harman, 1991*; *Niccoli and Partridge, 2012*). The mechanisms driving ageing are becoming increasingly well-understood, and conserved hallmarks of ageing, present in the aetiology of age-related diseases, have been described (*López-Otín et al., 2013*). Understanding how these physiological changes interact with each other, including which features are causative in age-related decline, represents a major challenge to the field (*López-Otín et al., 2013*; *Partridge et al., 2018*). Two prominent cellular processes identified as key players in organismal ageing are alteration of the epigenetic machinery and dysregulation of the insulin/Igf (IIS)/mechanistic target of rapamycin (mTOR) nutrient-sensing network (*Alic and Partridge, 2011*; *Benayoun et al., 2015*; *Johnson et al., 2013*; *López-Otín et al., 2013*; *Pal and Tyler, 2016*).

Alteration of the epigenetic machinery, including DNA methylation, post-translational modification of histones, and chromatin remodelling, can be driven by diverse stimuli during ageing (*Benayoun et al., 2015*). Multiple lines of evidence suggest that epigenetic alterations and perturbations can trigger progeroid syndromes or affect longevity in model organisms (*Pal and Tyler, 2016*; *Sen et al., 2016*). Enzymatic systems regulating epigenetic patterns, including DNA methylation and histone modifications, have been intensively studied. Beyond enzymatic regulation, there is growing

evidence that expression levels of histone proteins play a key role during the ageing process (*Benayoun et al., 2015*). Histone proteins pack and order genomic DNA into structural units called nucleosomes, and they constitute the major protein components of chromatin. Histones include the core histones H2A, H2B, H3, and H4, which form the nucleosome core, and the linked histone H1. Histone H3 protein levels decrease in aged yeast (*Feser et al., 2010*), the nematode worm *Caenorhabditis elegans* (*Ni et al., 2012*), and human senescent cells (*Ivanov et al., 2013*). Concordantly, over-expression of core histones H3 and H4 extended replicative lifespan in yeast, potentially attenuating the age-related loss of nucleosomes, transcriptional dysfunction, and genomic instability in aged yeast cells (*Feser et al., 2010*; *Hu et al., 2014*). Studies in yeast suggest that histone-driven loss of nucleosomes could contribute to ageing in other organisms, particularly given that histones have a high degree of structural and functional conservation in eukaryotes. However, almost nothing is known about the role of histone expression in longevity in multicellular organisms.

Dysregulation of the IIS/mTOR network at late ages also has substantial effects on organismal ageing (*López-Otín et al., 2013*). This network integrates multiple environmental inputs, including nutrient availability, to regulate metabolism, growth, stress resistance, immune responses, reproduction, and lifespan (*Alic and Partridge, 2011*; *Regan et al., 2020*; *Saxton and Sabatini, 2017*). Lowered activity of the IIS/mTOR network by nutritional, genetic, or pharmacological interventions can extend lifespan and reduce age-related pathologies in multiple organisms (*Fontana et al., 2010*; *Kenyon, 2010*; *Niccoli and Partridge, 2012*). Linkage studies of human longevity families and genome-wide association studies (GWAS) of populations suggest that the IIS/mTOR network is associated with longevity in humans (*Broer et al., 2015*; *Deelen et al., 2019*; *Johnson et al., 2015*; *Passtoors et al., 2013*; *Suh et al., 2008*).

mTOR is a serine/threonine protein kinase in the PI3K-related kinase family that forms two distinct protein complexes, mTOR Complex 1 (mTORC1) and 2 (mTORC2). Reduction of mTORC1 activity by genetic manipulation of key components of mTORC1, *TOR* or *Raptor*, extends lifespan in yeast, nematode worms *C. elegans*, the fruit fly *Drosophila melanogaster*, and mice (*Johnson et al., 2013*). The FDA-approved drug rapamycin directly targets mTORC1 and lowers its activity. Rapamycin treatment extends lifespan in diverse organisms, including mice, and attenuates a broad spectrum of age-related functional decline and diseases (*Johnson et al., 2013*; *Li et al., 2014*). In humans, rapamycin has been used clinically at high doses as an immunosuppressant to suppress tissue graft rejection, although these clinical doses are associated with negative metabolic side effects such as hyperglycaemia and insulin resistance. Recent studies, including those showing the beneficial effects of low-dose, short-term treatments with rapamycin analogs ('rapalogs') on response to vaccination in the elderly, without significant adverse side effects, suggest its therapeutic potential as a geroprotective compound (*Mannick et al., 2014*; *Mannick et al., 2018*; *Partridge et al., 2020*). Lifespan extension by rapamycin in *Drosophila* requires reduced S6K activity and increased autophagy downstream of mTORC1 (*Bjedov et al., 2010*). Consistently, genetic manipulations that reduce S6K activity (*Kapahi et al., 2004*; *Selman et al., 2009*) or activate autophagy (*Pyo et al., 2013*; *Ulgherait et al., 2014*) extend lifespan in both *Drosophila* and mice. More generally, activating expression of autophagy-related genes can prevent age-related dysfunction in a variety of tissues; for instance, limiting intestinal barrier dysfunction, memory impairment, and muscular dystrophy in animal models (*Hansen et al., 2018*). Given the promise of rapamycin and rapalogs to treat age-related decline in humans, understanding how the drug regulates autophagy, in which tissues, and how this leads to increased longevity, is crucial. This will allow for the development of more precise pharmacological treatments that circumvent unwanted side effects (*Arriola Apelo and Lamming, 2016*; *Li et al., 2014*).

Here we uncover an unexpected link between histone levels and mTORC1 signalling in *Drosophila* and mice. Rapamycin treatment increased expression of histone proteins through non-canonical eukaryotic initiation factor 3 (eIF3)-mediated translation in the intestine of *Drosophila*. Rapamycin treatment, or over-expression of histones H3 and H4, specifically in the enterocytes (ECs) of the fly intestine, caused chromatin rearrangement and heterochromatin relocation in EC nuclei. Increased expression of histones in ECs was a key step for rapamycin-dependent longevity and gut homoeostasis. Importantly, direct expression of H3/H4 in ECs was sufficient to extend lifespan and improve intestinal health during ageing. Increased expression of H3/H4 in ECs activated autophagy by epigenetic, transcriptional regulation of expression of autophagy-related genes, including Blue Cheese (Bchs), a selective autophagy cargo adaptor, which we demonstrated to be required and sufficient

for the effects of increased histone levels on intestinal autophagy, gut health, and lifespan. In mice, rapamycin treatment increased expression of histone proteins and the mammalian Bchs homolog *Wdfy3* transcript in the small intestines of aged individuals and altered the chromatin architecture in intestinal ECs, suggesting that the mTORC1-histone axis is at least partially conserved in mammals. Our findings unveil an mTORC1-histone axis as a crucial pro-longevity mechanism that can offer new directions for therapeutic anti-ageing interventions.

## Results

### Expression of core histones in the fly intestine is increased during ageing and by rapamycin treatment

To address a possible role for histones in the extension of lifespan induced by lowered mTORC1 activity in response to rapamycin treatment (*Bjedov et al., 2010*; *Figure 1A*; *Supplementary file 1*), we measured expression of histone proteins H3 and H4 during ageing in rapamycin-treated and control flies, in brain, muscle, fat body, and intestine (*Figure 1B*). In brain, muscle, and fat, neither rapamycin treatment nor age affected the expression of H3 or H4 protein (*Figure 1—figure supplement 1A–C*). In contrast, in the intestine rapamycin induced a marked increase in expression of both H3 and H4 proteins at all ages assessed, and there was also a slight increase in expression of these

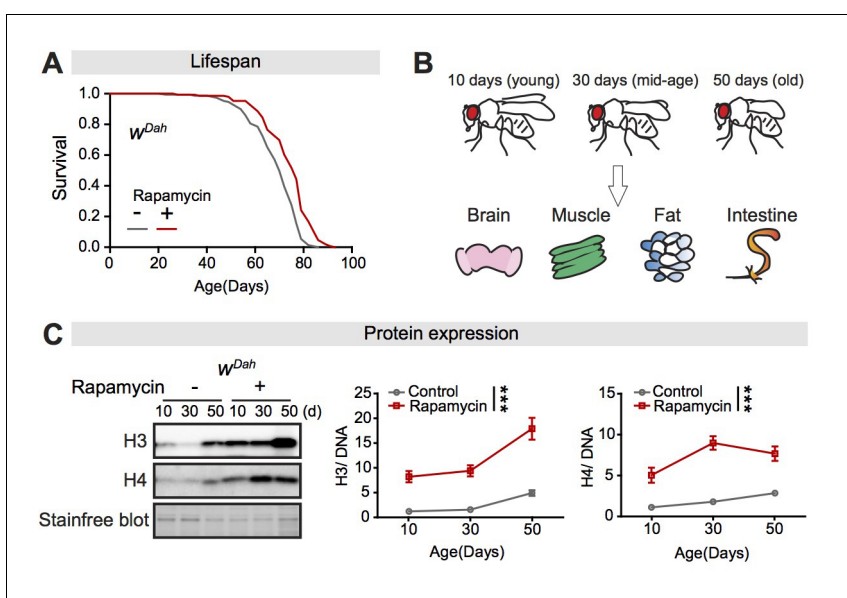

**Figure 1.** Expression of core histones in the fly intestine increases with age and in response to rapamycin treatment. (A) Adult-onset rapamycin treatment extended the lifespan of $w^{Dah}$ females (log-rank test, p=7.4E-08). See also *Supplementary file 1*. (B) Four tissues were dissected: brain, muscle, fat, and intestine, at 10 days, 30 days, and 50 days of adult age. (C) Expression of H3 and H4 in dissected intestines of $w^{Dah}$ controls significantly increased with age. Rapamycin substantially increased the expression of H3 and H4 in intestine (n = 4 biological replicates of 10 intestines per replicate, two-way ANOVA, H3 and H4, age p<0.05, treatment p<0.001, interaction p>0.05). The amount of protein was normalised to DNA, shown by stain-free blot.

The online version of this article includes the following source data and figure supplement(s) for figure 1:

**Source data 1.** Source data pertaining to *Figure 1*.
**Figure supplement 1.** Expression of core histones in different tissues and with rapamycin treatment.
**Figure supplement 1—source data 1.** Source data pertaining to *Figure 1—figure supplement 1*.
**Figure supplement 2.** Rapamycin increased expression of histone proteins without affecting their transcripts in the fly intestine.
**Figure supplement 2—source data 1.** Source data pertaining to *Figure 1—figure supplement 2*.
**Figure supplement 3.** Expression of core histones in the fly intestine increases with age but does not affect by dietary restriction (DR).
**Figure supplement 3—source data 1.** Source data pertaining to *Figure 1—figure supplement 3*.

proteins with age in control, untreated flies (*Figure 1C*). The intestine had much lower basal expression of H3 and H4 than did the other three tissues (*Figure 1—figure supplement 1D*). In the intestine, rapamycin increased expression of histone proteins by 2 days after the start of treatment (*Figure 1—figure supplement 2A*). The expression of core histones thus increased slightly during ageing in control flies and was strongly increased at all ages by rapamycin treatment, specifically in the intestine.

Previous studies have shown that dietary restriction (DR) has some similar effects on organismal physiology to rapamycin treatment (*Unnikrishnan et al., 2020*). We therefore tested if DR affected expression of histone proteins in the fly intestine. There was no difference in expression of H3 and H4 between intestines of flies fed control food and those fed food with a doubled yeast content (*Figure 1—figure supplement 3*). Increased histone protein expression was thus specific to treatment with rapamycin.

## Rapamycin treatment did not affect cell composition or EC polyploidisation in the intestine

The fly intestine contains four major cell types: intestinal stem cells (ISCs), which are mitotically active throughout the life course, multipotent enteroblasts, secretory enteroendocrine cells, and polyploid ECs that are the major differentiated cell type (*Lemaitre and Miguel-Aliaga, 2013*). The increase in expression of histone proteins in the intestine in response to rapamycin treatment could have been attributable to a change in cell composition or to the extent of polyploidisation of ECs. We therefore assessed the ratio of all cell types and of EC ploidy and found that neither was affected by rapamycin treatment (*Figure 2—figure supplement 1A–C*), suggesting that increased histone protein expression in response to rapamycin was not caused by changes to intestinal epithelial architecture or EC polyploidy.

Expression of core histones is increased in response to rapamycin treatment through eiF3 activity. mTORC1 is a central signalling hub that maintains cellular homeostasis through downstream effectors by transcriptional and post-transcriptional regulation (*Saxton and Sabatini, 2017*). To determine whether increased histone protein expression in response to rapamycin treatment was mediated transcriptionally, we measured expression of *histones H3* and *H4* transcripts in the intestines of flies treated with rapamycin. In young flies, up to day 10, transcript levels did not change in controls, and rapamycin treatment had no effect (*Figure 1—figure supplement 2B, C*). However, there was a marked age-related increase in controls at days 30 and 50, which was strongly attenuated by rapamycin treatment (*Figure 1—figure supplement 2C*). These results suggest that the age-related increase in histone protein levels may have been a consequence of increased transcript abundance, but that the rapamycin-dependent increase in histone H3 and H4 protein levels (*Figure 1C, Figure 1—figure supplement 1A*) was not and was instead mediated in a post-transcriptional manner through regulation of translation or protein stability.

We next tested whether rapamycin regulated histone protein levels through effects on their translation. Cycloheximide, which inhibits protein synthesis, abolished the increase in histone protein levels in response to rapamycin treatment (*Figure 2A*). This indicated that increased histone translation occurred in response to rapamycin treatment, which is not intuitive given that mTORC1 attenuation is known to suppress translation (*Saxton and Sabatini, 2017*). However, previous studies have demonstrated notable exceptions to this translational suppression, including histones, which can undergo increased translation via a non-canonical, eIF3-mediated mechanism (*Lee et al., 2015*; *Lee et al., 2016*; *Thoreen et al., 2012*). To test for the role of this mechanism, we knocked down expression of *eIF3d* or *eIF3g* in adult ECs by RNAi, which abolished the rapamycin-induced increased expression of histone proteins (*Figure 2B, C*), suggesting that the eIF3 protein complex was required. In addition, inhibiting the canonical mTORC1- eIF4 translation cascade, by knock-down of *eIF4e* in adult ECs by RNAi, recapitulated the rapamycin-induced increased expression of histone proteins (*Figure 2D*). This result was in line with the previous study showing that inhibition of eIF4 components can enforce mRNA translation through an eIF3-specialised pathway (*Lee et al., 2016*).

We examined whether rapamycin also regulated histone proteins through protein turnover. Neither perturbation of autophagy by ubiquitously reducing expression of *Atg5* by RNAi (*Bjedov et al., 2010*) nor inhibition of proteasome activity by treatment with bortezomib, a proteasome inhibitor (*Tain et al., 2017*), interfered with increased expression of histones in response to rapamycin

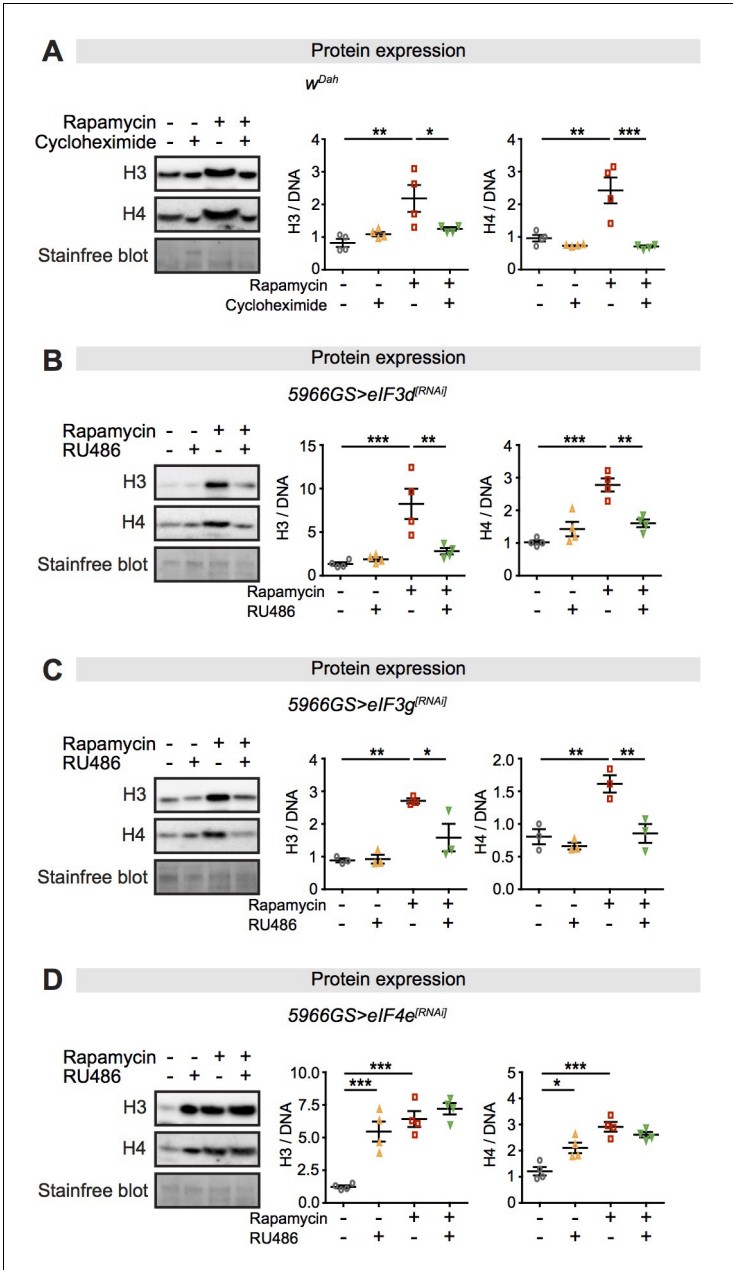

**Figure 2.** Expression of core histones in the fly intestine in response to rapamycin treatment and inhibition of translation or translation factors eukaryotic initiation factor (eIF)3 and eIF4. (A) Adult-onset cycloheximide treatment (1 mM) alone had no effect on histone expression but blocked increased expression of histones H3 and H4 in response to rapamycin treatment in intestines of flies at 2 days of age (n = 4 biological replicates of 10 intestines per replicate, two-way ANOVA, interaction, H3 $p<0.05$, H4 $p<0.01$; post-hoc test, *$p<0.05$, **$p<0.01$, ***$p<0.001$). (B, C) Adult-onset, enterocyte (EC)-specific knock-down of *eIF3d* or *eIF3g* by RNAi alone had no effect on histone expression but blocked increased expression of H3 and H4 in response to rapamycin treatment in intestine of flies at 20 days of age (n = 4 biological replicates of 10 intestines per replicate, two-way ANOVA, interaction, eIF3d RNAi H3 $p<0.01$, H4 $p<0.001$; eIF3g RNAi H3 $p<0.05$, H4 $p<0.05$; post-hoc test, *$p<0.05$, **$p<0.01$, ***$p<0.001$). (D) Adult-onset, EC-specific knock-down of *eIF4e* by RNAi alone increased expression of H3 and H4 to the same extent as did rapamycin treatment, with no additional effect of their combination in intestine of flies at 20 days of age (n = 4 biological replicates of 10 intestines per replicate, two-way ANOVA, interaction, H3 and H4 $p<0.01$; post-hoc test, *$p<0.05$, **$p<0.01$, ***$p<0.001$). The amount of protein was normalised to DNA, shown by stain-free blot.

The online version of this article includes the following source data and figure supplement(s) for figure 2:

*Figure 2 continued on next page*

*Figure 2 continued*

**Source data 1.** Source data pertaining to *Figure 2*.
**Figure supplement 1.** Rapamycin did not affect cell composition or enterocyte (EC) polyploidisation in the fly intestine.
**Figure supplement 1—source data 1.** Source data pertaining to *Figure 2—figure supplement 1*.
**Figure supplement 2.** Perturbation of autophagy or proteosome activity has no effect on expression of core histones in the fly intestine in response to rapamycin treatment.
**Figure supplement 2—source data 1.** Source data pertaining to *Figure 2—figure supplement 2*.

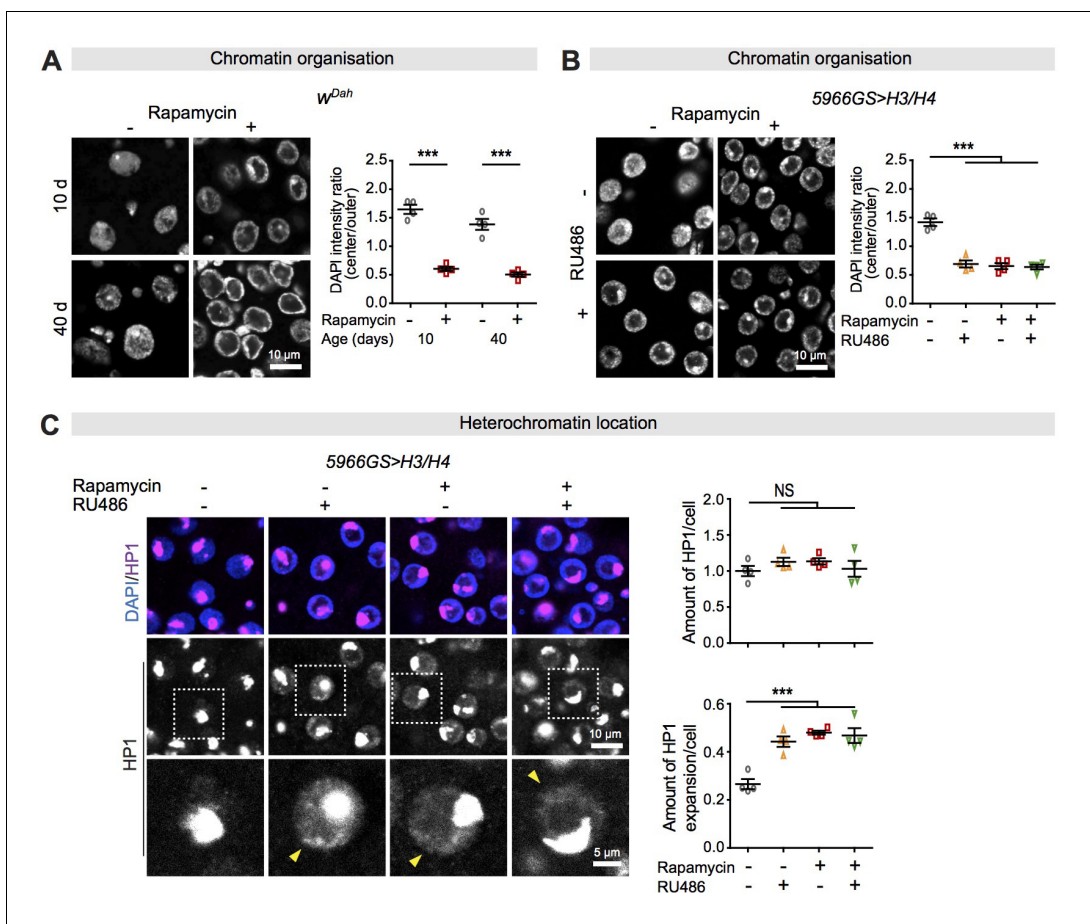

**Figure 3.** Increased histone expression in response to rapamycin treatment causes chromatin rearrangement and heterochromatin expansion across the nucleus in intestinal enterocytes (ECs). (**A**) Rapamycin induced a substantial accumulation of chromatin at the nuclear envelope in ECs (linear mixed model, interaction, p>0.05; post-hoc test, ***p<0.001). (**B**) Adult-onset, EC-specific expression of H3/H4 by the *5966GS* driver recapitulated the effect of rapamycin on the accumulation of chromatin at the nuclear envelope in intestine of flies at 20 days of age (linear mixed model, interaction, p<0.001; post-hoc test, ***p<0.001). (**C**) Adult-onset, EC-specific expression of H3/H4 by the *5966GS* driver had no effect on the total amount of HP1 in the presence or absence of rapamycin (linear mixed model, interaction, p>0.05; post-hoc test, NS p>0.05), but it altered the distribution of HP1 in the nucleus in the intestine of flies at 20 days of age (linear mixed model, interaction, p<0.001; post-hoc test, ***p<0.001). The yellow arrow indicates the expansion of HP1 to the whole nucleus. Each data point (n = 4 intestines) represents an average value from 3 to 5 ECs per intestine.

The online version of this article includes the following source data and figure supplement(s) for figure 3:

**Source data 1.** Source data pertaining to *Figure 3*.
**Figure supplement 1.** Increased histones are required for rapamycin-induced chromatin rearrangement and heterochromatin expansion in the fly intestine.
**Figure supplement 1—source data 1.** Source data pertaining to *Figure 3—figure supplement 1*.

(*Figure 2—figure supplement 2A, B*). Taken together, these results suggest that rapamycin mediated increased expression of histone proteins through translation factor eIF3.

## Increased expression of histones in ECs in response to rapamycin treatment alters chromatin architecture

Histones are basic proteins that help package genomic DNA to form chromatin. In yeast, loss of histones with age causes a decline in global nucleosome occupancy (*Hu et al., 2014*). Conversely, increased expression of histones can trigger a cytotoxic response to cytoplasmic-free histones (*Singh et al., 2010*) or result in an increase in the number of nucleosomes and altered chromatin structure (*Hu et al., 2014*). We observed that histone H3 remained in chromatin in intestines of both control and rapamycin-treated flies (*Figure 3—figure supplement 1A*), suggesting that rapamycin did not disturb histone incorporation into chromatin. We further examined whether the increase in expression of histones from rapamycin treatment resulted in altered chromatin structure. Micrococcal nuclease (MNase) cleaves and digests linker regions between nucleosomes, allowing the nucleosome number (occupancy) to be estimated. The number of mono-, di-, and tri-nucleosomes in the intestine of rapamycin-treated flies was substantially higher than in controls after a short (1 min) MNase digest. An extended digestion time led to the generation of more mononucleosomes from di- and tri-nucleosomes, revealing an even greater difference in mononucleosome number between rapamycin-treated and control intestines (*Figure 3—figure supplement 1B*). Over-expression of histones H3 and H4 in ECs elevated the number of nucleosomes in intestines as much as did rapamycin treatment, with no further increase in the combined treatment (*Figure 3—figure supplement 1B*). Thus, increased expression of histones resulted in increased nucleosome occupancy.

One consequence of increased nucleosome occupancy is a change in higher-order chromatin architecture (*Hauer and Gasser, 2017*; *Luger et al., 2012*). Interestingly, rapamycin treatment induced a substantial chromatin rearrangement in ECs, with marked accumulation of chromatin at the nuclear envelope in both young (10-day-old) and middle-aged (40-day-old) flies (*Figure 3A*). To determine if rapamycin induced this chromatin rearrangement by increasing histone expression, we either abolished increased histone expression by RNAi or directly overexpressed histones, and assessed the interaction with rapamycin treatment. Knock-down of either *histone H3* or *H4* in adult ECs by RNAi blocked the rapamycin-induced chromatin rearrangement (*Figure 3—figure supplement 1C, D*). Conversely, EC-specific over-expression of *H3* and *H4* recapitulated the effect of rapamycin treatment, with no further effect in the presence of rapamycin (*Figure 3B*). These results indicate that the increase in histone expression mediated the effect of rapamycin on chromatin arrangement.

Nucleosome occupancy and higher-order chromatin architecture eventually affect chromatin state. Heterochromatin is a tightly packed form of chromatin and is marked by heterochromatin protein 1 (HP1) (*Ebert et al., 2004*; *Grewal and Jia, 2007*). To investigate whether altered chromatin architecture led to heterochromatinisation in ECs, we examined the amount and the distribution of HP1 in EC nuclei. Rapamycin did not affect the amount of HP1, but it altered its distribution, by expanding it across the nucleus in ECs. Blocking-increased expression of *H3* or *H4* in response to rapamycin treatment did not affect the amount of HP1 but abolished HP1 expansion to the whole nucleus in response to rapamycin treatment (*Figure 3—figure supplement 1E, F*). Furthermore, over-expression of H3 and H4 in ECs recapitulated the effect of rapamycin treatment on this phenotype, with no additional effect in the presence of rapamycin (*Figure 3C*). Together, these results suggest that increased histone expression mediated the effects of rapamycin on higher-order chromatin architecture.

## Increased expression of histones in ECs mediates increased longevity and intestinal homeostasis in response to rapamycin

ECs play a key role in modulating ageing and age-related pathologies (*Bolukbasi et al., 2017*; *Guo et al., 2014*; *Lemaitre and Miguel-Aliaga, 2013*; *Resnik-Docampo et al., 2017*; *Salazar et al., 2018*). We therefore examined whether increased histone expression in ECs in the intestine mediated the effects of rapamycin on lifespan. Adult-onset knock-down of *H3* or *H4* by the *5966GS* driver alone had no effect on lifespan of control flies, but completely blocked the lifespan extension by rapamycin (*Figure 4A, B*; *Supplementary file 2*). Age-related intestinal pathologies are driven by

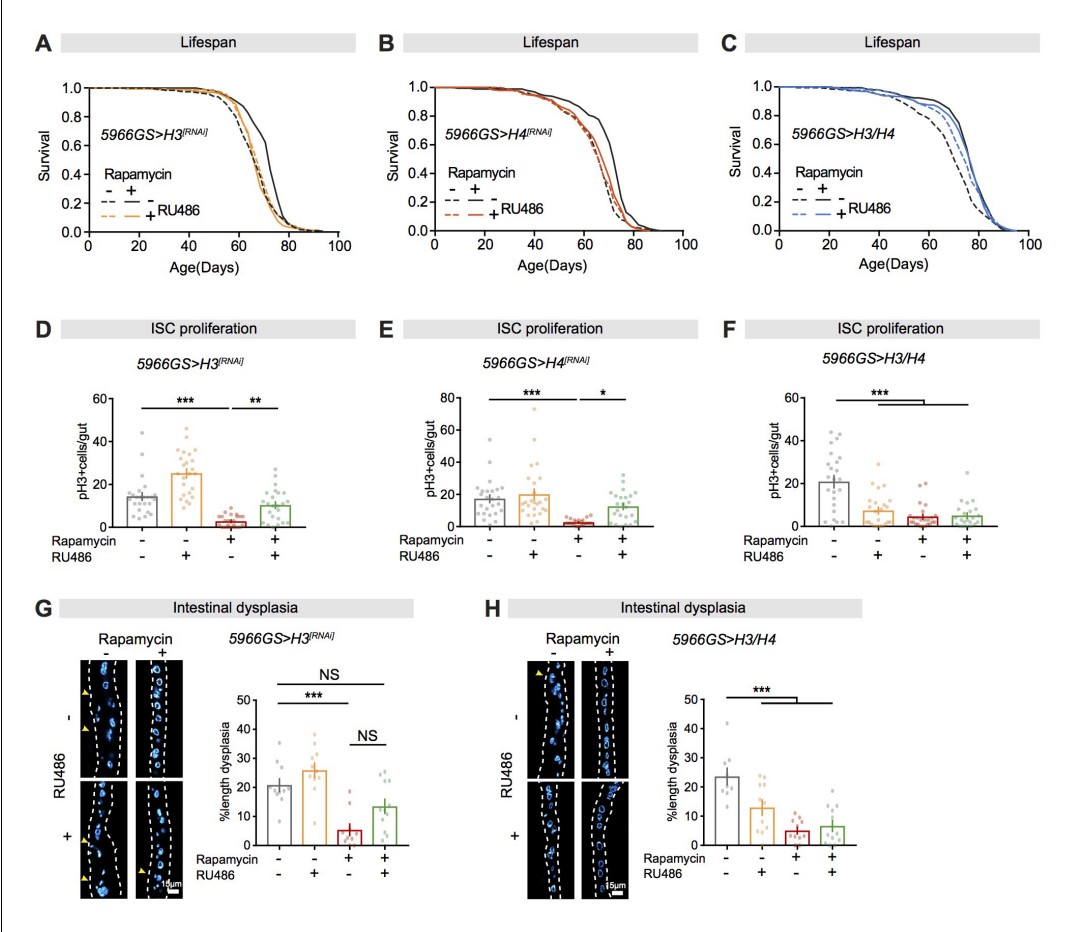

**Figure 4.** Increased histone expression in adult enterocytes (ECs) mediates lifespan extension and intestinal homeostasis from rapamycin treatment. (A, B) Rapamycin extended lifespan of control flies (log-rank test, *H3RNAi* p=3.80E-08; *H4RNAi* p=2.61E-12), but not of flies with knock-down of *H3 or H4* by RNAi in adult ECs (*H3RNAi* p=0.74; *H4RNAi* p=0.06). See also **Supplementary file 2**. (C) Adult-onset expression of H3/H4 in adult ECs extended lifespan (log-rank test, p=0.001) and had no additional effect on lifespan in the presence of rapamycin (Rapamycin vs. Rapamycin + RU, p=0.48). See also **Supplementary file 3**. (D, E) Knock-down of *H3 or H4* in adult ECs by RNAi counteracted the effects of rapamycin on intestinal stem cell (ISC) proliferation in flies at 20 days of age (n = 23–25 intestines, two-way ANOVA, interaction, p>0.05; post-hoc test, *p<0.05, **p<0.01, ***p<0.001). (F) Expression of H3/H4 in adult ECs reduced ISC proliferation in intestine of flies at 20 days of age (n = 23–24 intestines, two-way ANOVA, interaction, p<0.001; post-hoc test, ***p<0.001). (G) Knock-down of *H3* in adult ECs by RNAi partially blocked the effects of rapamycin on intestinal dysplasia in flies at 50 days of age (n = 10–12 intestines, two-way ANOVA, interaction, p>0.05; post-hoc test, NS p>0.05, ***p<0.001). (H) Expression of H3/H4 in adult ECs reduced intestinal dysplasia in 50-day-old flies (n = 9–12 intestines, two-way ANOVA, interaction, p<0.01; post-hoc test, ***p<0.001). The online version of this article includes the following source data for figure 4:

**Source data 1.** Source data pertaining to **Figure 4**.

both unregulated ISC division (**Biteau et al., 2008**; **Choi et al., 2008**) and loss of homeostasis in ECs (**Bolukbasi et al., 2017**; **Resnik-Docampo et al., 2017**; **Salazar et al., 2018**), both of which reduce lifespan. Rapamycin reduces age-associated ISC proliferation, attenuating intestinal dysplasia (**Fan et al., 2015**). To determine whether increased histone expression in ECs mediated the effects of rapamycin on intestinal homeostasis, we measured ISC proliferation. In line with the previous study (**Fan et al., 2015**), rapamycin treatment reduced pH3-positive cell number, a proxy for ISC proliferation (**Biteau et al., 2010**), in the intestine (**Figure 4D–F**). Knock-down of *H3 or H4* in adult ECs substantially attenuated the effect of rapamycin on ISC proliferation (**Figure 4D, E**) and on intestinal dysplasia in old flies (50-day-old) (**Figure 4G**). Increased longevity and intestinal homeostasis from rapamycin treatment thus both required the increased expression of histone proteins.

Reciprocally, over-expression of H3/H4 by the *5966GS* driver resulted in a marked extension of lifespan and did not further extend lifespan in rapamycin-treated flies (**Figure 4C**;

*Supplementary file 3*), suggesting that increased expression of histones in ECs mimicked the effects of rapamycin on lifespan. Furthermore, EC-specific expression of H3/H4 significantly attenuated ISC proliferation and intestinal dysplasia, while it had no further effect in rapamycin-treated flies (*Figure 4F–H*). Taken together, these results suggest that increased histone expression in ECs was sufficient to mediate the effects of rapamycin on longevity and gut health.

## Histones in ECs activate autophagy by mediating a transcriptional change upon rapamycin treatment

Changes in nucleosomes and chromatin mediate transcriptional responses, which can in turn affect ageing and health (*Hu et al., 2014*; *Larson et al., 2012*; *Sen et al., 2016*). To investigate whether these changes in ECs in response to rapamycin treatment were associated with changes in RNA expression, we compared RNA expression profiles of intestines of rapamycin-treated flies with controls. Rapamycin had a substantial impact on the entire transcriptome in the intestine, which increased with age (*Figure 5—figure supplement 1A*), with modest changes in gene expression at day 10, and substantial changes at days 30 and 50 (*Figure 5—figure supplement 1B*). Although we did not detect any significant enrichment of specific biological processes by Gene Ontology (GO) analysis, we noticed that expression of autophagy-related genes (e.g., *Bchs, Diabetes and obesity regulated (DOR), Stat92E, Atg4a,* and *Atg8a*) were affected by rapamycin treatment (*Figure 5—figure supplement 1C–E*). This is in line with previous studies showing that mTORC1 can influence expression of autophagy-related transcripts (*Di Malta et al., 2019*; *Martina et al., 2012*).

Autophagy plays an important role in gut health and longevity (*Hansen et al., 2018*). We therefore tested whether increased expression of histones in ECs could mediate transcriptional regulation of autophagy-related genes. Quantitative RT-PCR on RNA isolated from fly intestines showed that EC-specific knock-down of *H3* by RNAi abolished the effect of rapamycin on expression of the *Bchs* and *DOR* transcripts but not the *Stat92E* transcript (*Figure 5—figure supplement 2A*). Conversely, over-expression of H3 and H4 altered the expression of *Bchs* and *DOR* transcripts similarly to rapamycin treatment, with no additional effect of their combination (*Figure 5A*), suggesting that increased histone expression mediated increased expression of transcripts of autophagy-related genes *Bchs* and *DOR* in response to rapamycin.

Altered histone modifications (e.g., H3K9me3 and H3K27me3) regulate autophagy-related gene expression (*An et al., 2017*; *Wei et al., 2015*). We investigated whether increased histone levels affected the enrichments of histone modifications and HP1 on the *Bchs*, *DOR,* and *Stat92E* gene loci. ChIP-qPCR showed that expression of H3 and H4 altered the enrichment of H3K4me3, H3K9me3, H3K27me3, and HP1 on the *Bchs* and *DOR* gene loci, but not the *Stat92E* gene locus, similarly to rapamycin treatment, and with no additional effect of their combination (*Figure 5B*). Taken together, these results show that increased histone expression regulated expression of transcripts of autophagy-related genes *Bchs* and *DOR* through altering the enrichment of histone modifications (H3K4me3, H3K9me3, and H3K27me3) and HP1 on these gene loci in response to rapamycin.

Lowered mTORC1 activity can activate autophagy, either through transcriptional changes (*Martina et al., 2012*) or through mediating the phosphorylation status of Atg1, to regulate the activity of the Atg1/ULK1 autophagic complex and subsequent autophagic processes (*Jung et al., 2010*). To determine if increased histone levels induced autophagy by altering mTORC1 activity, we examined phospho-S6K levels, a direct output. As expected, rapamycin greatly decreased phospho-S6K levels, but EC-specific expression of H3/H4 did not affect phospho-S6K levels, in either the presence or the absence of rapamycin (*Figure 5—figure supplement 3A*). Furthermore, rapamycin treatment resulted in hyperphosphorylation of Atg1, shown by a slower-migrating band on western blot (*Figure 5—figure supplement 3B*), in line with previous studies (*Memisoglu et al., 2019*; *Yeh et al., 2010*). However, EC-specific expression of H3/H4 alone did not cause this effect (*Figure 5—figure supplement 3B*). Taken together, these results suggest that increased histone expression mediated autophagy through transcriptional change, rather than by affecting mTORC1 activity or phosphorylation status of Atg1.

The (macro)autophagy process is mediated by a number of autophagy-related proteins, which form double-membrane vesicles called autophagosomes that engulf cytoplastic material and subsequently fuse with lysosomes to form autolysosomes, where engulfed material is degraded (*Mizushima et al., 2010*). To determine the effect of increased histone expression on autophagy, we

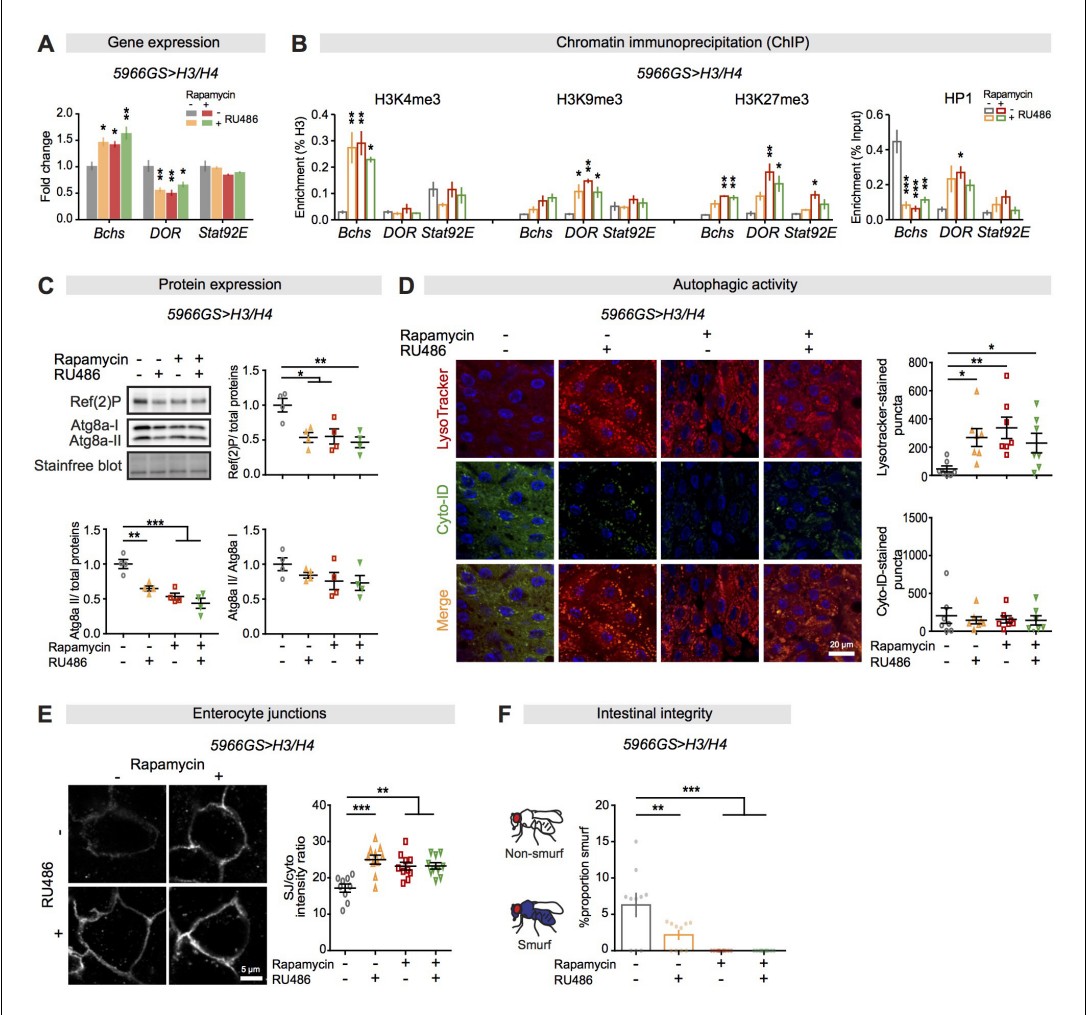

**Figure 5.** Increased histone expression in enterocytes from rapamycin treatment activates autophagy by altered histone marks and maintains gut barrier function. (A) Expression of H3/H4 in adult enterocytes (ECs) regulated expression of *Bchs* and *DOR* in the intestine of flies at 20 days of age (n = 4 biological replicates of 15 intestines per replicate, two-way ANOVA; post-hoc test, compared to controls, *p<0.05, **p<0.01). (B) Expression of H3/H4 in adult ECs mediated enrichment of H3K4me3, H3K9me3, H3K27me3, and HP1 on *Bchs* and *DOR* transcriptional start sites in the intestine of flies at 20 days of age (n = 3 biological replicates of 25 intestines per replicate, two-way ANOVA;post-hoc test, compared to controls, *p<0.05, **p<0.01, ***p<0.001). (C) Expression of H3/H4 in adult ECs decreased the amount of Atg8a-II and Ref(2)P (n = 4 biological replicates of 10 intestines per replicate, two-way ANOVA, interaction, p<0.05; post-hoc test, *p<0.05, **p<0.01, ***p<0.001). (D) Expression of H3/H4 in adult ECs substantially increased the number of LysoTracker-stained puncta and had no effect on the number of Cyto-ID-stained puncta in the intestine (n ≥ 6 intestines per condition; n = 2–3 pictures per intestine, data points represent the average value per intestine; linear mixed model, interaction, LysoTracker-stained puncta p<0.01; post-hoc test, *p<0.05, **p<0.01). (E) Expression of H3/H4 in adult ECs improved maintenance of coracle at septate junctions between ECs in the intestine of flies at 50 days of age. The ratio of septate junction (SJ)/cytoplasm fluorescence for coracle was high in the intestine of flies fed RU486 and/or rapamycin (n ≥ 9 intestines per condition; n = 3–5 ECs were observed per intestine, linear mixed model, interaction, p<0.01; post-hoc test, **p<0.01, ***p<0.001). (F) The number of Smurfs was significantly reduced in response to increased expression of H3/H4 in ECs and/or rapamycin at 60 days of age. Bar charts with n = 10 biological replicates of 15–20 flies per replicate (two-way ANOVA, interaction, p<0.05; post-hoc test, **p<0.01, ***p<0.001).

The online version of this article includes the following source data and figure supplement(s) for figure 5:

**Source data 1.** Source data pertaining to *Figure 5*.

**Figure supplement 1.** Rapamycin meditates a transcriptional response of autophagy-related genes in the fly intestine.

**Figure supplement 1—source data 1.** Source data pertaining to *Figure 5—figure supplement 1*.

**Figure supplement 2.** Increased histone expression in enterocytes is required for activation of autophagy and maintenance of gut barrier function from rapamycin treatment.

**Figure supplement 2—source data 1.** Source data pertaining to *Figure 5—figure supplement 2*.

**Figure supplement 3.** Increased histone expression does not affect mTORC1 activity.

**Figure supplement 3—source data 1.** Source data pertaining to *Figure 5—figure supplement 3*.

measured the levels of Atg8 and the *Drosophila* p62 homolog Ref(2)P. Atg8a-II, the active form of Atg8a, is a marker of autophagy, reflecting the number of autophagosomes (*Nagy et al., 2015*), while Ref(2)P is a cargo receptor for ubiquitinated proteins destined for degradation. Both are reduced upon persistently excessive autophagy (*Mizushima et al., 2010*). EC-specific expression of H3/H4 decreased the amount of Atg8a-II and Ref(2)P to the same degree as did rapamycin treatment, with no additional effect of their combination (*Figure 5C*), suggesting that increased histone expression mimicked the effect of rapamycin treatment on autophagy activation. To further assess the effect of increased histones on autophagy, we performed co-staining with LysoTracker, a fluorescent dye labelling acidic organelles, including autolysosomes, and Cyto-ID, a fluorescent dye labelling autophagosomes (*Oeste et al., 2013*). In line with a previous study (*Bjedov et al., 2010*), rapamycin treatment increased the number of LysoTracker-stained puncta in intestines (*Figure 5—figure supplement 2B*) while EC-specific knock-down of *H3* by RNAi abolished the increase (*Figure 5—figure supplement 2B*). Reciprocally, expression of H3/H4 increased the number of LysoTracker-stained puncta to the same extent as did rapamycin treatment, and neither treatment affected the number of Cyto-ID-stained puncta (*Figure 5D*), suggesting that expression of H3/H4 in ECs did not disturb autophagic flux. Together, these data suggest that increased expression of histones in ECs activated autophagy.

## Increased expression of histones in ECs improves gut barrier function

Activation of autophagy promotes increased intestinal junction and barrier integrity in worms and flies, and these play an important role in healthy longevity (*Hansen et al., 2018*). Rapamycin treatment attenuated the age-related loss of the bicellular junctional protein coracle (*Figure 5—figure supplement 2C*; *Resnik-Docampo et al., 2017*; *Salazar et al., 2018*). EC-specific knock-down of *H3* by RNAi abolished the effect of rapamycin on maintenance of coracle levels at EC junctions, while expression of H3/H4 resulted in maintenance of coracle similarly to rapamycin treatment, without further effect of their combination (*Figure 5E*). These results suggest that increased expression of histones in response to rapamycin treatment led to better junction maintenance in the intestine of old flies. To further investigate whether activation of autophagy improved intestinal barrier integrity in old flies, we fed aged flies with a blue dye that normally does not leak out of the intestine into the body and scored the number of flies with extra-intestinal accumulation of the blue dye (the 'Smurf' phenotype; *Clark et al., 2015*; *Rera et al., 2012*). Rapamycin treatment resulted in a reduction of barrier function loss, and this effect was abolished by knock-down of *H3* in adult ECs (*Figure 5—figure supplement 2D*). Expression of H3/H4 in ECs resulted in a modest, but significant, reduction in the number of Smurf flies and had no further effect in the presence of rapamycin (*Figure 5F*). Taken together, these results suggest that increased histone expression in ECs in response to rapamycin treatment improved the maintenance of EC junctions and overall intestinal integrity in old flies, which may in turn have promoted systemic health and increased lifespan.

## Autophagy is required downstream of the mTORC1-histone axis for increased health and survival

Autophagy activation is necessary for lifespan extension in response to rapamycin in flies (*Bjedov et al., 2010*). To elucidate whether autophagy activation mediates the effects of increased histone expression in ECs on lifespan and intestinal homeostasis, we inhibited autophagy in ECs by knock-down of *Atg5* expression by RNAi (*Bjedov et al., 2010*). Reduction of Atg5 substantially reduced the number of LysoTracker-stained puncta following increased expression of H3/H4 in ECs (*Figure 6A*), suggesting that Atg5 was required for the effect of increased histone expression on autophagy activation. We next examined whether autophagy activation was required for the beneficial effects of increased histone expression in ECs on survival and intestinal health. EC-specific knock-down of *Atg5* alone did not affect lifespan, but it abolished the increase in response to increased expression of H3/H4 (*Figure 6B*; *Supplementary file 4*). Furthermore, EC-specific knock-down of *Atg5* completely blocked the effects of increased expression of H3/H4 on intestinal dysplasia and maintenance of gut integrity (*Figure 6C, D*). Interestingly, we obtained similar results by EC-specific knock-down of expression of *Atg1,* a key gene with multiple roles in autophagy, including in autophagy initiation, through its phosphorylation, and in autophagosome formation and/or fusion with lysosomes (*Kraft et al., 2012*; *Nakatogawa et al., 2012*; *Noda and Fujioka, 2015*). Knock-

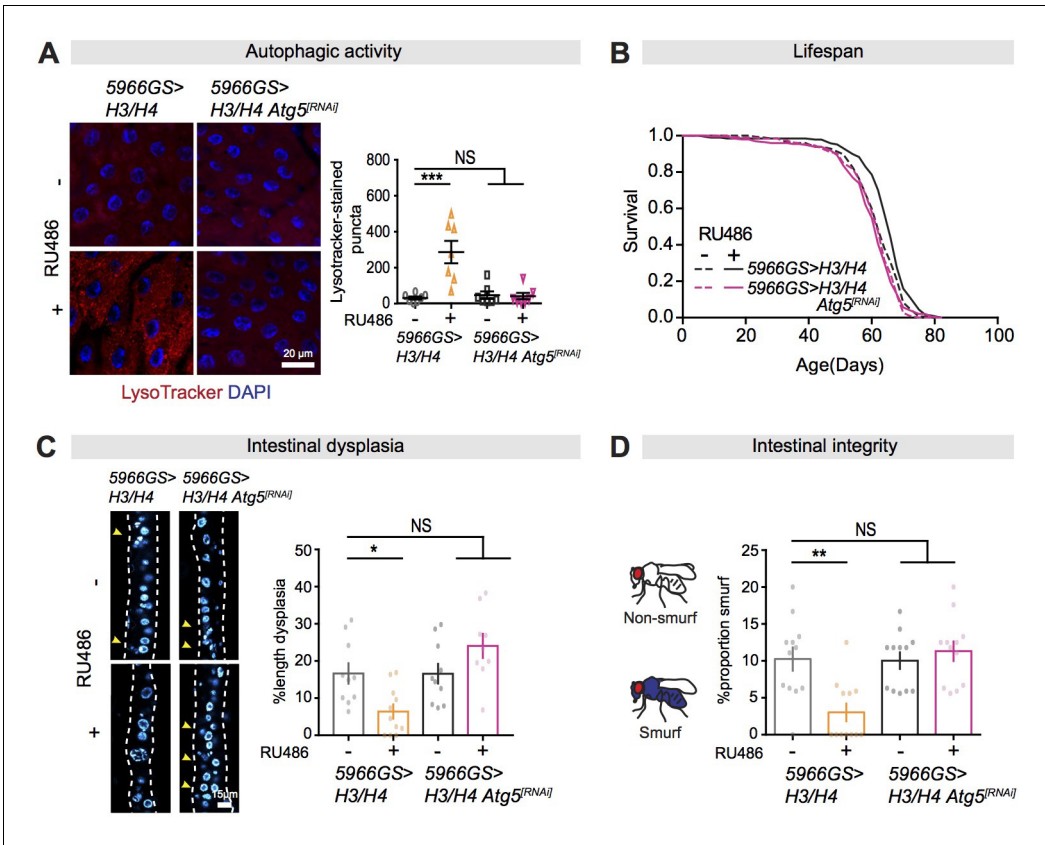

**Figure 6.** Autophagy activation is necessary for mTORC1-histone axis on survival and intestinal homeostasis. (**A**) Knock-down of *Atg5* abolished the effect of expression of H3/H4 in enterocytes (ECs) on induction of LysoTracker-stained puncta in the intestine of flies at 20 days of age (n ≥ 6 intestines per condition; n = 2–3 images per intestine, data points represent the average value per intestine; linear mixed model, interaction, p<0.001; post-hoc test, ***p<0.001). (**B**) Knock-down of *Atg5* abolished the increase in lifespan in response to expression of H3/H4 in adult ECs. *5966GS>H3/H4* females showed increased lifespan in the presence of RU486 (log-rank test, p=0.0001), but *5966GS>H3/H4 Atg5[RNAi]* females did not (p=0.49). See also *Supplementary file 4*. (**C**) Knock-down of *Atg5* blocked the effect of expression of H3/H4 in adult ECs on intestinal dysplasia at 50 days of age (n = 9–12 intestines, two-way ANOVA, interaction, p<0.01; post-hoc test, NS p>0.05, ***p<0.001). (**D**) Knock-down of *Atg5* abolished the effects of expression of H3/H4 in adult ECs on gut integrity at 60 days of age. Bar charts with n = 12 biological replicates of 15–20 flies per replicate (two-way ANOVA, interaction, p<0.01; post-hoc test, **p<0.01, ***p<0.001).

The online version of this article includes the following source data and figure supplement(s) for figure 6:

**Source data 1.** Source data pertaining to *Figure 6*.

**Figure supplement 1.** Autophagy activation is required for mTORC1-histone axis on survival and intestinal homeostasis.

**Figure supplement 1—source data 1.** Source data pertaining to *Figure 6—figure supplement 1*.

down of *Atg1* inhibited the increase in autophagy in response to over-expression of H3/H4 (*Figure 6—figure supplement 1A*) and blocked the beneficial effects of increased expression of H3/H4 on gut health (*Figure 6—figure supplement 1B*). Together, these results suggest that increased autophagy is required for the beneficial effects of increased histone expression in response to rapamycin treatment for the increases in gut health and longevity.

## The selective autophagy cargo adaptor Bchs mediates the effects of rapamycin and histones on the intestine and lifespan

Autophagy not only functions as a bulk degradation pathway, but also contributes to selective clearance of unwanted cellular material, including aggregated proteins, damaged mitochondria, and

invading pathogens (*Zaffagnini and Martens, 2016*). WDFY3 is a cargo adaptor for selective degradation of ubiquitinated protein aggregates and physically interacts with Atg5 and p62 (*Clausen et al., 2010*; *Filimonenko et al., 2010*). Mutants in the *Drosophila Wdfy3* homolog *Bchs* show shortened lifespan and neurodegeneration (*Finley et al., 2003*; *Sim et al., 2019*). Given that the expression of *Bchs* was increased in response to rapamycin treatment or over-expression of H3/H4 in ECs, we examined if Bchs was required for the effects of these treatments on the intestine and lifespan. Reduction of Bchs expression by RNAi in combination with either rapamycin treatment or over-expression of H3/H4 in ECs blocked the increase of LysoTracker-stained puncta (*Figure 7A, Figure 7—figure supplement 1A*) in response to H3/H4, suggesting that Bchs was crucial for the effects of increased histone expression on autophagy activation. Knock-down of *Bchs* alone had no effect on lifespan, but it abolished the effects of both rapamycin treatment and histone over-expression on lifespan (*Figure 7B, Figure 7—figure supplement 1B*; *Supplementary files 5* and *6*). It also abolished the effects of these treatments on intestinal dysplasia and gut integrity (*Figure 7C, D*, *Figure 7—figure supplement 1C, D*). Conversely, EC-specific over-expression of *Bchs* was sufficient to recapitulate the effects of these treatments on autophagy, lifespan, and intestinal homoeostasis (*Figure 7E–H*; *Supplementary file 7*). Moreover, we found that neither knocking down nor over-expressing *Bchs* in ECs influenced mTORC1-mediated phosphorylation of Atg1 (*Figure 7—figure supplement 2A, B*). Taken together, these data suggest that Bchs is a required target for the effects of increased expression of histones on autophagy and longevity, and acts independently of mTORC1-mediated phosphorylation of Atg1.

## Rapamycin treatment increases expression of histones and alters the chromatin structure in the small intestine of mice

There are many physiological and functional similarities between the fly and mammalian intestine, especially the signalling pathways that regulate intestinal regeneration and disease (*Apidianakis and Rahme, 2011*; *Jiang and Edgar, 2012*). To investigate whether the mTORC1-histone axis is conserved between fly and mammals, we examined whether rapamycin increased the expression of histones in small intestines in mice. Expression of all of the core histones (H2A, H2B, H3, and H4) in the small intestine of female mice was significantly increased by rapamycin treatment at 12 months and 22 months of age (*Figure 8A, B*), consistent with our results from flies. In mammals, intestinal villi are small projections that extend into the lumen of the small intestine, and they are predominantly composed of ECs (*Sancho et al., 2015*). Rapamycin treatment induced a modest, but significant, chromatin rearrangement in epithelial cells in villi, with marked accumulation of chromatin at the nuclear envelope in cells of rapamycin-fed mice at 12 months and 22 months of age (*Figure 8C*). Rapamycin treatment also increased nucleosome occupancy in 22-month-old rapamycin-fed mouse intestines (*Figure 8D*). Furthermore, expression of *Wdfy3* transcript in the small intestine in 22-month-old mice increased in response to rapamycin treatment (*Figure 8E*). Taken together, these results suggest that the mTORC1-histone axis may respond to mTORC1 inhibition in similar ways in flies and mammals.

## Discussion

Changes in histone expression levels during ageing is a common phenomenon in diverse organisms (*Benayoun et al., 2015*). In yeast, over-expression of histones H3 and H4 prevents age-related nucleosome loss and transcriptional dysfunction, and extends replicative lifespan (*Feser et al., 2010*; *Hu et al., 2014*) Therefore, it is important to understand how histones contribute to longevity and in which tissues of multicellular organisms they play such a role. In this study, we have shown that histones H3 and H4 act downstream of mTORC1 to play a critical role in gut ECs in mediating autophagy to promote intestinal health and lifespan extension.

The expression of histones dynamically responds to cellular and environmental stresses in order to alter nuclear architecture, both to protect genomic DNA from damage and to orchestrate transcriptional programmes (*Feser et al., 2010*; *Matilainen et al., 2017*; *Maze et al., 2015*). Both nutrient-sensing pathways and chromatin regulation, including that mediated by histones, affect longevity, and perturbations to either of them can cause age-associated pathologies (*López-Otín et al., 2013*). However, it is unknown whether these processes act together to affect the ageing process. Here, we focused on females because their lifespan is increased much more than is that of

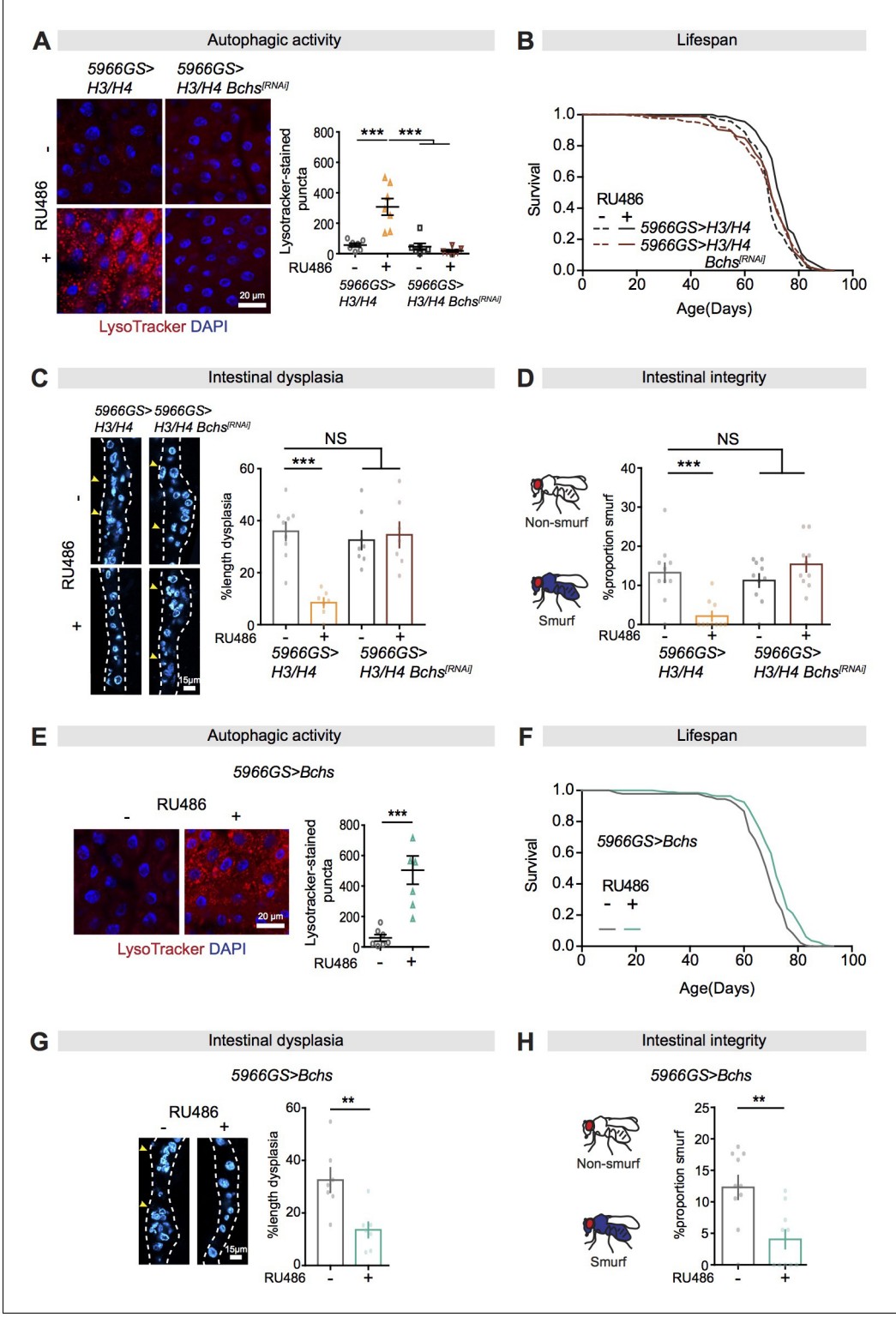

**Figure 7.** Bchs is a required target for autophagy activation, lifespan extension, and intestinal homeostasis from the mTORC1-histone axis. (**A**) Knock-down of *Bchs* abolished the effect of expression of H3/H4 in enterocytes (ECs) on induction of LysoTracker-stained puncta in the intestine of flies at 20 days of age (n = 7 intestines per condition; n = 3 images per intestine, data points represent the average value per intestine; linear mixed model, interaction, p<0.001; post-hoc test, ***p<0.001). (**B**) Knock-down of *Bchs* abolished the effects of expression of H3/H4 in adult ECs on lifespan. *5966GS>H3/H4* females showed increased lifespan in the presence of RU486 (log-rank test, p=7.55E-08), but *5966GS>H3/H4 Bchs[RNAi]* females did not (p=0.90). See also *Figure 7 continued on next page*

*Figure 7 continued*

*Supplementary file 5*. (C) Knock-down of *Bchs* blocked the effect of expression of H3/H4 in adult ECs on intestinal dysplasia at 50 days of age (n = 7–9 intestines, two-way ANOVA, interaction, p<0.001; post-hoc test, ***p<0.001). (D) Knock-down of *Bchs* abolished the beneficial effects of expression of H3/H4 in adult ECs on gut integrity at 60 days of age. Bar charts with n = 10 biological replicates of 15–20 flies per replicate (two-way ANOVA, interaction, p<0.001; post-hoc test, **p<0.01, ***p<0.001). (E, F) Expression of *Bchs* in adult ECs substantially increased the number of LysoTracker-stained puncta in the intestine (n = 7 intestines per condition; n = 3 images per intestine, data points represent the average value per intestine; linear mixed model, ***p<0.001) and extended lifespan (log-rank test, p=4.92E-06). See also *Supplementary file 7*. (G) Expression of *Bchs* in adult ECs reduced intestinal dysplasia in 50-day-old flies (n = 7 intestines, Student's t test, **p<0.01). (H) The proportion of Smurfs at 60 days of age was significantly reduced in response to increased expression of *Bchs* in ECs and/or rapamycin treatment. Bar charts with n = 10 biological replicates of 15–20 flies per replicate (Student's t test, **p<0.01).

The online version of this article includes the following source data and figure supplement(s) for figure 7:

**Source data 1.** Source data pertaining to *Figure 7*.

**Figure supplement 1.** Bchs is a required autophagic target for rapamycin-induced lifespan extension and intestinal homeostasis.

**Figure supplement 1—source data 1.** Source data pertaining to *Figure 7—figure supplement 1*.

**Figure supplement 2.** Manipulated Bchs expression does not influence mTORC1-dependent phosphorylation of Atg1.

males upon rapamycin treatment (*Bjedov et al., 2010*) and they show age-related intestinal decline that is attenuated by rapamycin treatment (*Fan et al., 2015*; *Regan et al., 2016*). Our findings reveal an interaction between mTORC1 signalling and histones, which determines longevity. Lowered mTORC1 activity by rapamycin treatment caused increased expression of histones in the intestine in *Drosophila* and mice, and changes in nuclear architecture of ECs and transcription of autophagy-related genes. Interestingly, the basal protein expression level of histones was substantially higher in brain, muscle, and fat than in intestine in *Drosophila*, and rapamycin did not further increase histone levels in these three tissues, possibly because their chromatin is already fully occupied by histones. Our findings therefore elucidate a novel intestine-specific mechanism connecting nutrient-sensing pathways and histone-driven chromatin alterations in ageing, which can be regulated by mTORC1 attenuation through rapamycin treatment.

The *Drosophila* intestine consists of four main cell types that have distinct physiological functions and genomic DNA content. Our findings show that increased expression of core histones in response to rapamycin treatment was not caused by either cell composition change or EC polyploid-isation, and the drop in ISC proliferation may instead reflect increased health and persistence of ECs. Given the crucial role of histone proteins in packaging genomic DNA into nucleosomes to form chromatin, it is essential to finely regulate histone levels in the cell. In line with a previous study demonstrating that the expression of histone transcripts and proteins is uncoupled in aged yeast (*Feser et al., 2010*), we found that in *Drosophila* ECs lowered activity of mTORC1 by rapamycin treatment-elevated histone protein expression, independent of the abundance of histone transcripts. In mice also, the increase in lifespan from rapamycin treatment can be dissociated from the reduction in global translational activity (*Garelick et al., 2013*). Previous studies suggest that histones are exceptions to translation suppression upon mTOR attenuation, with their translational efficiencies increased through translation factor eIF3 (*Lee et al., 2015*; *Lee et al., 2016*; *Thoreen et al., 2012*). Here, we reveal that increased histones in the fly intestine in response to rapamycin treatment is regulated through translation, specifically via the activity of eIF3 in ECs. Together, these findings suggest that regulation of expression of specific protein subsets, including histones, is a key effector for rapamycin-induced longevity.

Global histone loss accompanied with nucleosome reduction occurs in aged budding yeast, and over-expression of H3/H4 ameliorates age-related nucleosome loss and extends replicative lifespan (*Hu et al., 2014*). Although we did not observe age-related histone loss in the *Drosophila* or mouse tissues that we examined, increased histone expression from rapamycin treatment, or EC-specific expression of H3/H4, caused the number of nucleosomes to increase. Furthermore, this resulted in a higher-order chromatin structural rearrangement in intestinal ECs. Importantly, our findings show this chromatin rearrangement did not happen over ageing, possibly because the ageing-induced

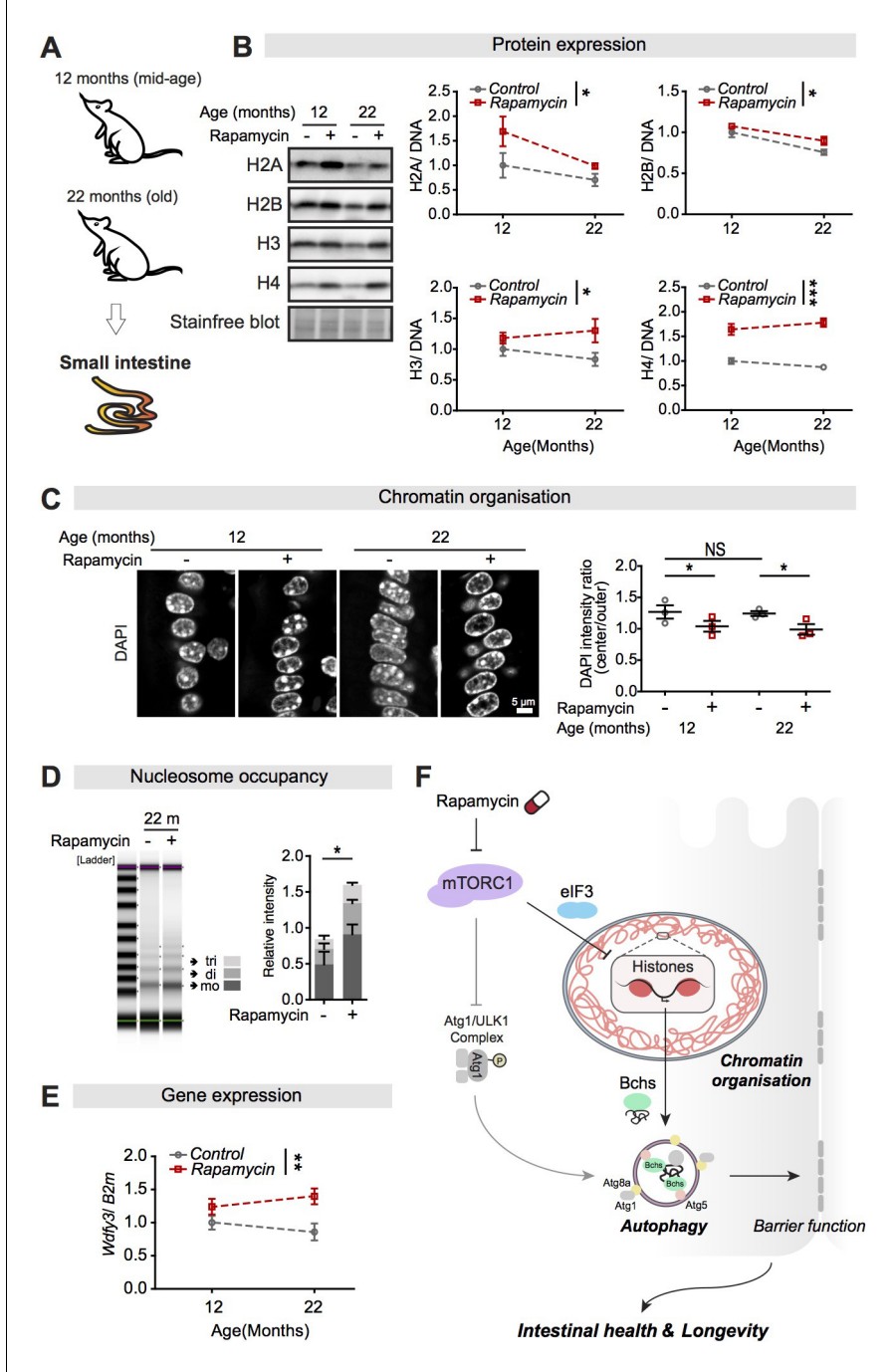

**Figure 8.** Rapamycin treatment upregulates expression of histones and *Wdfy3* transcript, alters chromatin structure, and increases the number of nucleosomes in the small intestine of mice. (A) Female mice were sacrificed at 12 months and 22 months of age, and the jejunum of the small intestine was dissected. (B) Rapamycin substantially increased expression of H2A, H2B, H3, and H4 compared to controls in the small intestine of mice (n = 3 jejunums, two-way ANOVA; treatment *p<0.05; ***p<0.001). The amount of protein was normalised to DNA, shown by stain-free blot. (C) Rapamycin induced a substantial accumulation of chromatin at the nuclear envelope in cells in villi of the small intestine of mice at 12 months and 22 months of age (n = 3 jejunums per condition; n = 40–45 cells were observed per intestine, linear mixed model; post-hoc test, NS p>0.05, *p<0.05). (D) The number of nucleosomes in the intestine increased markedly in response to rapamycin treatment in mice at 22 months of age. Gel electrophoresis of 5 min micrococcal nuclease (MNase) digestions showed that the majority of nucleosomes after digestion were trinucleosomes (tri), dinucleosomes (di), and mononucleosomes (mo). The number of nucleosomes was normalised to input (0 min) (n = 3 jejunums, two-way ANOVA, post-hoc test,

*Figure 8 continued on next page*

*Figure 8 continued*

*p<0.05). (E) Rapamycin substantially increased *Wdfy3* in the small intestine of mice compared to controls at 22 months of age (n = 3 jejunums, mean ± SEM, two-way ANOVA; treatment **p<0.01). (F) Model of the relationship linking mTORC1, histones, autophagy, and longevity.

The online version of this article includes the following source data for figure 8:

**Source data 1.** Source data pertaining to *Figure 8*.

increase of histone proteins was subtle and much lower than that induced by rapamycin treatment. Chromatin organisation plays an essential role in cellular senescence and organismal ageing. For instance, profound chromatin change has been reported in senescent fibroblasts, including the formation of senescence-associated heterochromatin foci (SAHF) (*Chandra et al., 2015*; *Chandra et al., 2012*), and these changes to chromatin structure can directly affect transcriptional programmes (*Finlan et al., 2008*; *Zuin et al., 2014*). Regulation of histone expression levels in ECs hence may be important for mediating their transcriptional programme.

Interestingly, we found that increased histone expression in ECs led to activation of autophagy in the fly intestine, accompanied by attenuation of age-related intestinal pathologies and extension of lifespan. Autophagy plays a crucial role in a number of conserved longevity paradigms, including reduced IIS/mTOR network and DR in multiple organisms (*Hansen et al., 2018*). Furthermore, genetically inducing autophagy globally, or activating selective autophagy mechanisms, extends lifespan in worms (*Kumsta et al., 2019*), flies (*Aparicio et al., 2019*; *Ulgherait et al., 2014*), and mice (*Pyo et al., 2013*). Generally, autophagy is considered to be regulated by mTORC1 by altering phosphorylation status of the Atg1/ULK1 complex (*Jung et al., 2010*). However, autophagy can be also controlled by epigenetic and transcriptional mechanisms, and several lines of evidence suggest that epigenetic regulation of autophagy-related genes activates autophagy, and is key for somatic homeostasis (*Füllgrabe et al., 2016*; *Lapierre et al., 2015*). In line with these previous studies, we found that the increased histones activated autophagy by altering enrichment of H3K4me3, H3K9me3, H3K27me3, and HP1 at the loci of autophagy-related genes, including Bchs, a selective autophagy cargo adaptor, to mediate their transcriptional expression and activate autophagy without affecting mTORC1 activity or phosphorylation status of Atg1.

The age-related decline of structure and function in the intestine has been shown to lead to intestinal pathologies and mortality (*Regan et al., 2016*; *Rera et al., 2012*; *Resnik-Docampo et al., 2017*; *Salazar et al., 2018*). Given the importance of the intestine for health and longevity, it is crucial to preserve its structure and function during ageing. We demonstrate that the histone protein levels in intestinal ECs mediate intestinal health and longevity in response to rapamycin treatment. Importantly, over-expressing histones H3/H4 in adult ECs recapitulated the effects of rapamycin treatment, which attenuated age-related structural and functional decline in the intestine and extended lifespan. Consistent with several previous studies showing that activation of autophagy promotes maintenance of cell-cell junctions and barrier function in the intestine (*Hansen et al., 2018*), we found that activation of autophagy was required for, and is sufficient to recapitulate, the effects on barrier integrity by increased levels of histones in ECs.

Atg1 has multiple functions in autophagy process. While its phosphorylation is essential for autophagy initiation (*Jung et al., 2010*), its protein (i.e., AIM/LIR sequence) can interact with Atg8a and therefore contributes to autophagosome formation and/or fusion with lysosomes (*Kraft et al., 2012*; *Nakatogawa et al., 2012*; *Noda and Fujioka, 2015*). In line with these findings, we found that hyperphosphorylation of Atg1, which is induced by rapamycin, was unaffected by increased histones H3/H4. Instead Atg1 protein functioned downstream of increased histones in ECs. Increased transcription of Bchs, which was sufficient to mediate autophagy (*Sim et al., 2019*), was a key downstream effector of histone-induced intestinal health and longevity.

In sum, the simplest model to integrate the role of rapamycin, histones, and autophagy in extension of lifespan and preservation intestinal health is presented in *Figure 8F*. We propose that lowered mTORC1 activity by rapamycin increases expression of histone proteins in intestinal ECs in a post-transcriptional manner through the activity of eIF3. This increased expression of histones in ECs alters chromatin architecture and transcriptional output in ECs, including of autophagy-related genes that activate intestinal autophagy, resulting in preserved gut health and extended lifespan. This mTORC1-histone axis can activate autophagy via epigenetic and transcriptional regulation of Bchs

which subsequently works together with other autophagy-related proteins, for example, Atg1, Atg5, and Atg8a, bypassing the canonical mTORC1-mediated phosphorylation of Atg1 autophagy initiation.

Importantly, we found that the effects of rapamycin treatment on histone protein levels, *Wdfy3* transcript, and chromatin architecture were conserved in mice. Rapamycin treatment increased expression of all core histones, nucleosome occupancy, and expression of *Wdfy3* transcript in the small intestines of mice and altered higher-order chromatin structure in intestinal villi cells. Several lines of evidence from previous studies have suggested that rapamycin affects histone methylation and chromatin states in aged mice (*Gong et al., 2015*; *Wang et al., 2017*). Furthermore, in humans, rapamycin affects chromatin organisation in fibroblasts from normal individuals in a way that mimics that seen in fibroblasts from centenarians (*Lattanzi et al., 2014*), further supporting the idea that the mTORC1-histone axis is a pro-longevity mechanism in mammals. Our study highlights the mTORC1-histone axis as a novel, pharmacological target that requires further investigation for its potential role in geroprotection.

## Materials and methods

### Fly husbandry

The wildtype *D. melanogaster* stock, *Dahomey,* was collected in 1970 in Dahomey (now Benin), and since then it has been maintained in large population cages with overlapping generations on a 12L:12D cycle at 25°C. The white *Dahomey* ($w^{Dah}$) stock was derived by incorporation of the *white* gene deletion from $w^{1118}$ into the outbred *Dahomey* background by successive backcrossing. All mutants were backcrossed for at least six generations into the wild type, $w^{Dah}$, maintained in population cages. Stocks were maintained and experiments conducted at 25°C on a 12 hr:12 hr light/dark cycle at 60% humidity, on food (1× SYA) containing 10% (w/v) brewer's yeast, 5% (w/v) sucrose, and 1.5% (w/v) agar unless otherwise noted. The following stocks were used in this study and are listed in the Key resources table. *UAS-H3/H4* strain was generated by combining the *UAS-H3* and *UAS-H4* strains. *UAS-H3* strain was generated by cloning the H3 cDNA into the pUAST attb vector. pUAST attb H3 was inserted into the fly genome by the φC31 and attP/attB integration system using the attP40 landing site.

### Mouse husbandry

Female mice of the genetically heterogeneous UM-HET3 stock (CByB6F1 × C3D2F1) were used in this study. They were bred, housed, and given ad libitum access to normal or rapamycin-containing chow under specific pathogen-free conditions. Rapamycin was added to the food at concentration of 14 ppm (mg of drug per kg of food). Mice were fasted for 18 hr before euthanasia at the age of 12 months and 22 months, and small intestines were dissected into different parts, including duodenum, jejunum, and ileum, then snap-frozen in liquid nitrogen and embedded in paraffin. The jejunum part was used in this study. The mouse work was approved by the University of Michigan's Institutional Committee on the Use and Care of Animals.

### Lifespan assay

For lifespan assays and all other experiments, flies were reared at standard density before being used for experiments. Crosses were set up in cages with grape juice agar plates. Embryos were collected in PBS and dosed into bottles at 20 μl per bottle to achieve standard density. The flies were collected over a 24 hr period and allowed 48 hr to mate after eclosing as adults. Flies were subsequently lightly anaesthetised with $CO_2$, and females were sorted into vials. RU486 (Sigma) and/or rapamycin (LC Laboratories) dissolved in ethanol was added to food at appropriate concentrations (RU486 100 μM, rapamycin 200 μM). For control food, ethanol alone was added. Flies were maintained continuously on the appropriate food.

### Cycloheximide/bortezomib treatment

Cycloheximide (Sigma) or bortezomib (Sigma) dissolved in ethanol was added to food at appropriate concentrations (cycloheximide 1 mM, bortezomib 2 μM) with or without rapamycin. For control food,

ethanol alone was added. Flies were kept continuously on the appropriate food until being dissected.

## Gut barrier assay ('Smurf' assay)

Flies were aged on standard 1× SYA food and then switched to SYA food containing 2.5% (w/v) Brilliant Blue FCF (Sigma). Flies were examined after 48 hr, as previously described (*Martins et al., 2018*; *Regan et al., 2016*; *Rera et al., 2012*).

## RNA isolation and quantitative RT-PCR

Tissue of female flies was dissected, frozen on dry ice, and stored at −80°C. Total RNA from guts of 10 flies was extracted using TRIzol (Invitrogen) according to the manufacturer's instructions. mRNA was reverse transcribed using random hexamers and the SuperScript III First Strand system (Invitrogen). Quantitative PCR was performed using Power SYBR Green PCR (Applied Biosystems) on a QuantStudio 6 instrument (Applied Biosystems) by following the manufacturer's instructions. Primers used are listed in the Key resources table.

## Immunoblotting

Female fly tissues were homogenised in 100 μl 1× RIPA Lysis and Extraction Buffer (Thermo Fisher) containing PhosSTOP (Roche) and cOmplete, Mini, EDTA-free Protease Inhibitor Cocktail (Roche). Extracts were cleared by centrifugation, protein content determined by using Pierce BCA Protein Assay (Thermo Fisher), and DNA content determined by using Qubit dsDNA HS Assay (Invitrogen). Approximately 10 μg of protein extract or 100 ng of DNA extract was loaded per lane on polyacrylamide gel (4–20% Criterion, Bio-Rad). Proteins were separated and transferred to PVDF membrane. HRP-conjugated secondary antibodies (Invitrogen) were used. Blots were developed using the ECL detection system (Amersham). Immunoblots were analysed using Image Lab program (Bio-Rad laboratories).

## Subcellular isolation

Fly guts were homogenised in 100 μl 1% Triton X-100 lysis buffer containing PhosSTOP (Roche) and cOmplete, Mini, EDTA-free Protease Inhibitor Cocktail (Roche), then centrifuged for 15 min at 4°C. The supernatant contains cytoplasmic and necleoplasmic faction, and the pellet contains chromatin faction.

## MNase assay

Fly guts were homogenised in 200 μl Nuclei Prep buffer (Zymo Research). Extracts were pelleted by centrifugation, resuspended in 120 μl MN Digestion buffer (Zymo Research), and DNA content determined by using Qubit dsDNA HS Assay (Invitrogen). Approximately 50 ng of DNA extract was used for enzymatic treatment. DNA was digested using 0.0025 U MNase. Treatment was stopped at different time points (1, 2, 5, 10 min). Nucleosomal DNA purification was done by following the manufacturer's instructions. DNA fragments were analysed using High Sensitivity D5000 ScreenTape (Agilent Technologies) in a 4200 TapeStation instrument (Agilent Technologies).

## Chromatin immunoprecipitation (ChIP)

Guts were dissected in PBS and immediately cross-linked in 1% formaldehyde for 10 min, fixation was subsequently stopped with 0.125 M glycine and washed in PBS, centrifuged at 4°C. Pellets were homogenised in Lysis buffer, centrifuged at 4°C, suspended in Shearing buffer, and sonicated by Covaris M220 sonicator. The following antibodies for immunoprecipitation were used: anti-histone H3 (Abcam #ab1791), anti-H3K4me3 (Abcam #ab8580), anti-H3K9me3 (Abcam #ab8898), and anti-HP1 (DSHB #C1A9). The pre-immune serum was used as mock control. Enrichment after IP was measured relative to input with qPCR. Primers used are listed in the Key resources table.

## Cyto-ID and LysoTracker staining, imaging, and image analysis

Cyto-ID staining selectively labels autophagic vacuoles, and LysoTracker dye accumulates in low pH vacuoles, including lysosomes and autolysomes. Combination of both gives a better assessment of the entire autophagic process (*Oeste et al., 2013*). For the dual staining, complete guts were

dissected in PBS and stained with Cyto-ID (Enzo Life Sciences, 1:1000) for 30 min, then stained with LysoTracker Red DND-99 (Thermo Fisher, 1:2000) with Hoechst 33342 (1 mg/ml, 1:1000) for 3 min. For the experiment only with LysoTracker staining, guts were stained with LysoTracker Red and Hoechst 33342 directly after dissection. Guts were mounted in Vectashield (Vector Laboratories, H-1000) immediately. Imaging was performed immediately using a Leica TCS SP8 confocal micro-scope with a 20× objective plus 5× digital zoom in. Three separate images were obtained from each gut. Settings were kept constant between images. Images were analysed by Imaris 9 (Bitplane).

## Immunohistochemistry and imaging of the *Drosophila* intestine

The following antibodies were used for immunohistochemistry of fly guts. Primary antibodies: anti-PH3 (Cell Signaling #9701, 1:200), anti-Lamin C (DSHB #LC28.26, 1:250), anti-HP1 (DSHB #C1A9, 1:500), anti-Coracle (DSHB #C615.16, 1:100), and anti-Prospero (DSHB #MR1A, 1:250). Secondary antibodies: Alexa Flour 488 goat anti-mouse (A11001, 1:1000) and Alexa Flour 594 goat anti-rabbit (A11012, 1:1000). Guts were dissected in PBS and immediately fixed in 4% formaldehyde for 30 min, and subsequently washed in 0.1% Triton-X/PBS (PBST), blocked in 5% BSA/PBST, incubated in pri-mary antibody overnight at 4°C, and in secondary antibody for 1 hr at room temperature (RT). Guts were mounted in Vectashield, scored, and imaged as described above. For dysplasia measurement, the percentage of intestinal length was blind-scored from luminal sections of the R2 region of intestines.

## Immunohistochemistry and imaging of the mouse intestine

Staining was performed on 5-μm-thick sections of formalin-fixed paraffin-embedded (FFPE) jejunum samples of 12- and 22-month-old rapamycin-treated and control animals. Deparaffinised, heat-medi-ated antigen retrieval with 10 mM sodium citrate buffer (pH 6) and blocking with IHC blocking buffer (5% FBS, 2.5% BSA in 1× PBS) were carried out according to standard protocols. Primary antibody incubations were performed overnight at 4°C in reaction buffer (0,25% BSA, 5% FBS, 2 g NaCl, and 0.1 g Triton X-100 in 1× PBS) using the primary antibody Lamin A/C (CST #2032, 1:50). Secondary antibody incubations were performed 1 hr at RT using Alexa Flour 594 goat anti-rabbit (A11012, 1:400), followed by washing and DAPI staining (1 μg/μl). Samples were washed in PBS 0.5% Triton or PBS and mounted in Vectashield (Vector Laboratories H-1000).

## Library preparation and RNA sequencing

For transcriptomic analysis, guts were dissected from control and rapamycin-treated females at the age of 10 days, 30 days, and 50 days. Total RNA was extracted from 25 guts (three replicates) using Trizol (Thermo Fisher) following standard protocols. DNA concentrations were evaluated using a Qubit 2.0 fluorometer (Life Technologies) before DNase I treatment (Thermo Fisher). After adjusting final RNA concentration to 100 ng/μl, 2–3 μl ERCC ExFold RNA Spike-In Mixes (Life Technologies) was added for normalisation to the DNA content of the sample. Ribosomal RNA depletion libraries were generated at the Max Planck Genome Centre Cologne (MPGCC). RNA sequencing was per-formed with an Illumina HighSeq2500 with 150 bp read length read at MPGCC. At least 37.5 million single-end reads were obtained for each sample.

## RNA sequencing data analysis

Raw sequence reads were quality-trimmed using Flexbar (v2.5.0) and aligned using HiSat (v2.0.14) against the Dm6 reference genome (*Dodt et al., 2012*; *Kim et al., 2015*). Mapped reads were fil-tered using SAMtools (v1.2) (*Li et al., 2009*), and guided transcriptome assembly was done using StringTie (v1.04) (*Pertea et al., 2015*). Merging of assembled transcriptomes and differential gene expression was performed using deseq2 analysis after ERCC normalised. The data are accessible through (GEO: GSE148002).

## Quantification and statistical analysis

Statistical analyses were performed in Prism (GraphPad) or R (version 3.5.5) except for log-rank test using Excel (Microsoft). For the quantification of the chromatin arrangement, Leica LAS X-3D (Leica) was used to measure the fluorescence intensity of the DAPI and Lamin C staining. For the quantifica-tion of the total amount of HP1, Fiji was used to measure the sum of fluorescence intensity from the

nucleus, and the amount of HP1 per cell in all treatments was compared to controls. The amount of HP1 in peripheral location in nucleus was divided by the total amount of HP1 to obtain the amount of HP1 expansion. Sample sizes and statistical tests used are indicated in the figure legends, and Tukey post-hoc test was applied to multiple comparisons correction. Error bars are shown as standard error of the mean (SEM). The criteria for significance are NS (not significant) $p > 0.05$; $*p < 0.05$; $**p < 0.01$, and $***p < 0.001$.

## Acknowledgements

We thank Christian Kukat and the FACS and Imaging Core Facility at the Max Planck Institute for Biology of Ageing for their help with microscopy data. We gratefully acknowledge Julia Hoffmann and the Bioinformatics Core Facility, including Jorge Boucas, Sven Templer, and Franziska Metge at the Max Planck Institute for Biology of Ageing for their help with data analysis and the Max Planck Genome Center Cologne for generation of sequencing libraries and performing next-generation sequencing. We gratefully acknowledge Michelle Dassen, Jenny Fröhlich, and Paula Juricic for help in preparing tissues. We are grateful to Prof. Jun Hee Lee for providing us with *Drosophila* Atg1 antibody and Prof. Péter Nagy for providing us with *Drosophila* Atg8a antibody. We thank Luke Tain, Martin Graef, and Peter Tessarz for useful discussions. The Bloomington *Drosophila* Stock Center (NIH P40OD018537) and Vienna *Drosophila* Resource Center (VDRC) are acknowledged for fly lines. This project has received funding from the European Research Council (ERC) under the European Union's Horizon 2020 research and innovation programme no. 741989 and the Max-Planck-Gesellschaft. Yu-Xuan Lu was supported by an EMBO Long-Term Fellowship (ALTF 419-2014). Mouse experiments were supported by the Glenn Foundation for Medical Research.

## Additional information

### Funding

| Funder | Grant reference number | Author |
|---|---|---|
| Horizon 2020 Framework Programme | 741989 | Linda Partridge |
| European Molecular Biology Organization | ALTF419-2014 | Yu-Xuan Lu |
| Glenn Foundation for Medical Research | | Richard A Miller |
| Max Planck Institute for Biology of Ageing | Open-access funding | Linda Partridge |

The funders had no role in study design, data collection and interpretation, or the decision to submit the work for publication.

### Author contributions

Yu-Xuan Lu, Conceptualization, Formal analysis, Funding acquisition, Investigation, Writing - original draft, Project administration; Jennifer C Regan, Formal analysis, Writing - review and editing; Jacqueline Eßer, Lisa F Drews, Thomas Weinseis, Julia Stinn, Oliver Hahn, Investigation; Richard A Miller, Resources, Funding acquisition; Sebastian Grönke, Formal analysis; Linda Partridge, Conceptualization, Supervision, Funding acquisition, Writing - review and editing

### Author ORCIDs

Yu-Xuan Lu https://orcid.org/0000-0001-6751-5250
Jennifer C Regan https://orcid.org/0000-0003-2164-9151
Sebastian Grönke http://orcid.org/0000-0002-1539-5346
Linda Partridge https://orcid.org/0000-0001-9615-0094

### Ethics

Animal experimentation: The work on mice at Michigan was reviewed and approved by the Institutional Animal Care and Use Committee. The original protocol was PRO00008130, approved February 13, 2018. This was renewed as Protocol PRO00009981 on December 7, 2020.

### Decision letter and Author response

Decision letter https://doi.org/10.7554/eLife.62233.sa1
Author response https://doi.org/10.7554/eLife.62233.sa2

## Additional files

### Supplementary files

• Supplementary file 1. Inhibition of mTORC1 activity by rapamycin treatment extends lifespan in females. Related to *Figure 1*.

• Supplementary file 2. Knock-down of *histone H3* or *H4* in adult enterocytes blocks rapamycin-induced lifespan extension. Related to *Figure 4*.

• Supplementary file 3. Over-expression of H3/H4 in enterocytes recapitulates rapamycin-induced lifespan extension. Related to *Figure 4*.

• Supplementary file 4. Knock-down of *Atg5* associated with the expression of H3/H4 in enterocytes abolished the benefits of increased histones on lifespan extension. Related to *Figure 6*.

• Supplementary file 5. Knock-down of *Bchs* associated with the expression of H3/H4 in enterocytes abolished the benefits of increased histones on lifespan extension. Related to *Figure 7*.

• Supplementary file 6. Knock-down of *Bchs* in enterocytes abolished rapamycin-induced lifespan extension. Related to *Figure 7—figure supplement 1*.

• Supplementary file 7. Over-expression of Bchs in enterocytes extends lifespan in females. Related to *Figure 7*.

• Transparent reporting form

### Data availability

Sequencing data have been deposited in GEO under accession code GSE148002.

The following dataset was generated:

| Author(s) | Year | Dataset title | Dataset URL | Database and Identifier |
|---|---|---|---|---|
| Lu Y, Regan JC, Eßer J, Drews LF, Weinseis T, Stinn J, Hahn O, Miller RA, Grönke S, Partridge L | 2021 | A TORC1/histone axis regulates chromatin organization and non-canonical induction of autophagy to ameliorate ageing | https://www.ncbi.nlm.nih.gov/geo/query/acc.cgi?acc=GSE148002 | NCBI Gene Expression Omnibus, GSE148002 |

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

# Appendix 1

**Appendix 1—key resources table**

| Reagent type (species) or resource | Designation | Source or reference | Identifiers | Additional information |
|---|---|---|---|---|
| Antibody | Histone H2A antibody (rabbit polyclonal) | Active Motif | Cat# 39209 RRID:AB_2793184 | WB (1:1000) |
| Antibody | Anti-histone H2B antibody – ChIP Grade (mouse monoclonal) | Abcam | Cat# ab52484 RRID:AB_1139809 | WB (1:10,000) |
| Antibody | Anti-histone H3 antibody – ChIP Grade (rabbit polyclonal) | Abcam | Cat# ab1791 RRID:AB_302613 | WB (1:10,000) ChIP (2 µg) |
| Antibody | Histone H4 antibody (pAb) (rabbit polyclonal) | Active Motif | Cat# 39269 RRID:AB_2636967 | WB (1:3000) |
| Antibody | Lamin C (mouse monoclonal) | Developmental Studies Hybridoma Bank | Cat# LC28.26 RRID:AB_528339 | WB (1:3000) |
| Antibody | Atg1 (rabbit polyclonal) | From Jun Hee Lee's lab, USA | | WB (1:2000) |
| Antibody | Atg8a (rabbit polyclonal) | From Péter Nagy's lab, Hungary | | WB (1:3000) |
| Antibody | Phospho-*Drosophila* p70 S6 Kinase (Thr398) (rabbit polyclonal) | Cell Signaling | Cat#9209 RRID:AB_2269804 | WB (1:1000) |
| Antibody | Total S6K (rabbit polyclonal) | Self-made | | WB (1:1000) |
| Antibody | Anti-histone H3 (trimethyl K4) antibody – ChIP Grade (rabbit polyclonal) | Abcam | Cat# ab8580 RRID:AB_306649 | ChIP (2 µg) |
| Antibody | Anti-histone H3 (trimethyl K9) antibody – ChIP Grade (rabbit polyclonal) | Abcam | Cat# ab8898 RRID:AB_306848 | ChIP (2 µg) |
| Antibody | Anti-trimethyl-histone H3 (Lys27) Antibody (rabbit polyclonal) | Merck Millipore | Cat# 07-449 RRID:AB_310624 | ChIP (2 µg) |
| Antibody | HP1 (mouse monoclonal) | Developmental Studies Hybridoma Bank | Cat# C1A9 RRID:AB_528276 | IF (1:500) |
| Antibody | Mouse (G3A1) mAb IgG1 Isotype Control (mouse monoclonal) | Cell Signaling | Cat# 5415 RRID:AB_10829607 | ChIP (2 µg) |
| Antibody | Rabbit IgG, polyclonal - Isotype Control (rabbit polyclonal) | Abcam | Cat# ab171870 RRID:AB_2687657 | ChIP (2 µg) |
| Antibody | Coracle (mouse monoclonal) | Developmental Studies Hybridoma Bank | Cat# C615.16 RRID:AB_1161644 | IF (1:100) |
| Antibody | Phospho-Histone H3 (Ser10) (rabbit polyclonal) | Cell Signaling | Cat# 9701 RRID:AB_331535 | IF (1:500) |

*Continued on next page*

*Appendix 1—key resources table continued*

| Reagent type (species) or resource | Designation | Source or reference | Identifiers | Additional information |
|---|---|---|---|---|
| Antibody | Lamin A/C (rabbit polyclonal) | Cell Signaling | Cat# 2032 RRID:AB_2136278 | IF (1:500) |
| Antibody | Prospero (mouse monoclonal) | DSHB | Cat# MR1A RRID:AB_528440 | IF (1:200) |
| Antibody | Goat anti-Mouse IgG (H+L) Cross-Adsorbed Secondary Antibody, Alexa Fluor 488 | Thermo Fisher Scientific | Cat# A11001 RRID:AB_2534069 | IF (1:1000) |
| Antibody | Goat anti-Rabbit IgG (H+L) Cross-Adsorbed Secondary Antibody, Alexa Fluor 594 | Thermo Fisher Scientific | Cat# A11012 RRID:AB_2534079 | IF (1:1000) |
| Chemical compound, drug | Rapamycin | LC Laboratories | Cat# R-5000 | |
| Chemical compound, drug | RU486 (Mifepristone) | Sigma | Cat# M8046 | |
| Chemical compound, drug | Cycloheximide | Sigma | Cat# 239763 | |
| Chemical compound, drug | Bortezomib | Sigma | Cat# 5043140001 | |
| Chemical compound, drug | Brilliant Blue FCF | Sigma | Cat# 80717 | |
| Commercial assay or kit | TRIzol Reagent | Invitrogen | Cat#15596018 | |
| Commercial assay or kit | SuperScript III First Strand Master Mix | Invitrogen | Cat#15596018 | |
| Commercial assay or kit | High Sensitivity D5000 ScreenTape | Agilent Technologies | Cat# 5067-5593 | |
| Commercial assay or kit | VECTASHIELD Antifade Mounting Medium | Vector Laboratories | Cat# H-1000 RRID:AB_2336789 | |
| Commercial assay or kit | RIPA Lysis and Extraction Buffer | Thermo Scientific | Cat# 89901 | |
| Commercial assay or kit | 4–20% Criterion TGX Stain-Free Protein Gel, 26 well, 15 µl | Bio-Rad | Cat# 5678095 | |
| Commercial assay or kit | Amersham Hybond P Western blotting membranes, PVDF | GE Healthcare Amersham | Cat# GE10600023 | |
| Commercial assay or kit | PhosSTOP | Roche | Cat# 04906837001 | |
| Commercial assay or kit | cOmplete, Mini, EDTA-free Protease Inhibitor Cocktail | Roche | Cat# 11836170001 | |
| Commercial assay or kit | Qubit dsDNA HS Assay Kit | Invitrogen | Cat# Q32854 | |
| Commercial assay or kit | Qubit dsRNA HS Assay Kit | Invitrogen | Cat# Q32852 | |
| Commercial assay or kit | Pierce BCA Protein Assay Kit | Thermo Scientific | Cat# 23227 | |
| Commercial assay or kit | Power SYBR Green PCR Master Mix | Applied Biosystems | Cat# 4367659 | |
| Commercial assay or kit | TaqMan Gene Expression Master Mix | Applied Biosystems | Cat# 4369016 | |
| Commercial assay or kit | EZ Nucleosomal DNA Prep Kit | Zymo Research | Cat# D5220 | |

*Continued on next page*

*Appendix 1—key resources table continued*

| Reagent type (species) or resource | Designation | Source or reference | Identifiers | Additional information |
|---|---|---|---|---|
| Commercial assay or kit | LysoTracker Red DND-99 | Invitrogen | Cat# L7528 | |
| Commercial assay or kit | Cyto-ID Autophagy detection kit 2.0 | Enzo Life Sciences | Cat# ENZ-KIT175 | |
| Strain, strain background (*Drosophila melanogaster*) | *w^Dah* | This lab | | |
| Genetic reagent (*D. melanogaster*) | *DaGS* | Bloomington *Drosophila* Stock Center | 8641 | |
| Genetic reagent (*D. melanogaster*) | *5966GS* | *Guo et al., 2014* | | |
| Genetic reagent (*D. melanogaster*) | *esgGal4* | Kyoto Stock Center | 104863 | |
| Genetic reagent (*D. melanogaster*) | *UAS-H3^[RNAi]* | Vienna *Drosophila* Resource Center | KK109374 | |
| Genetic reagent (*D. melanogaster*) | *UAS-H4^[RNAi]* | Vienna *Drosophila* Resource Center | KK109059 | |
| Genetic reagent (*D. melanogaster*) | *UAS-Bchs^[RNAi]* | Vienna *Drosophila* Resource Center | KK110785 | |
| Genetic reagent (*D. melanogaster*) | *UAS-Bchs.HA* | Bloomington *Drosophila* Stock Center | 51636 | |
| Genetic reagent (*D. melanogaster*) | *UAS-Atg5^[RNAi]* | *Scott et al., 2004*; *Ren et al., 2009* | | |
| Genetic reagent (*D. melanogaster*) | *UAS-Atg1^[RNAi]* | Bloomington *Drosophila* Stock Center | 26731 | |
| Genetic reagent (*D. melanogaster*) | *UAS-eIF3d^[RNAi]* | Vienna *Drosophila* Resource Center | 330545 | |
| Genetic reagent (*D. melanogaster*) | *UAS-eIF3g^[RNAi]* | Vienna *Drosophila* Resource Center | GD28937 | |
| Genetic reagent (*D. melanogaster*) | *UAS-eIF4e^[RNAi]* | Vienna *Drosophila* Resource Center | GD7800 | |
| Genetic reagent (*D. melanogaster*) | *UAS-H3/H4* | This lab | | |
| Genetic reagent (*D. melanogaster*) | *UAS-mCD8::GFP.L* | Bloomington *Drosophila* Stock Center | 5137 | |
| Strain, strain background (mouse) | UM-HET3 stock (CByB6F1 × C3D2F1) | This lab | | |
| Sequence-based reagent | Primer for Q-RT-PCR Act5C_F1: AGGC CAACCGTGAGAAGATG | This paper | | |
| Sequence-based reagent | Primer for Q-RT-PCR Act5C_R1: GGGG AAGGGCATAACCCTC | This paper | | |
| Sequence-based reagent | Primer for Q-RT-PCR His3_F1: CCACGCA AACAACTGGCTAC | This paper | | |
| Sequence-based reagent | Primer for Q-RT-PCR His3_R1: TGCGGAT TAGAAGCTCGGTG | This paper | | |

*Continued on next page*

*Appendix 1—key resources table continued*

| Reagent type (species) or resource | Designation | Source or reference | Identifiers | Additional information |
|---|---|---|---|---|
| Sequence-based reagent | Primer for Q-RT-PCR His4_F1: CGGATAGCAGGCTTCGTGAT | This paper | | |
| Sequence-based reagent | Primer for Q-RT-PCR His4_R1: GGTCGTGGTAAAGGAGGCAA | This paper | | |
| Sequence-based reagent | Primer for Q-RT-PCR Bchs_F1: AGCCTCACCACGCTAAAGAAG | This paper | | |
| Sequence-based reagent | Primer for Q-RT-PCR Bchs_R1: CTCATGTCGTTTGACGGACAG | This paper | | |
| Sequence-based reagent | Primer for Q-RT-PCR DOR_F1: CTTGATCTCGGGGTGTCGAC | This paper | | |
| Sequence-based reagent | Primer for Q-RT-PCR DOR_R1: CTTCAACTGTACGGCCGCAT | This paper | | |
| Sequence-based reagent | Primer for Q-RT-PCR Stat92_F1: AAGCTGCTTGCCCAAAACTAC | This paper | | |
| Sequence-based reagent | Primer for Q-RT-PCR Stat92_R1: GACGCATTGTGAGTACGATGG | This paper | | |
| Sequence-based reagent | Primer for Q-RT-PCR B2m_Mouse_F1: TTCTGGTGCTTGTCTCACTGA | This paper | | |
| Sequence-based reagent | Primer for Q-RT-PCR B2m_Mouse_R1: CAGTATGTTCGGCTTCCCATTC | This paper | | |
| Sequence-based reagent | Primer for Q-RT-PCR Wdfy3_Mouse_F1: GAGGCTCTGGAGTGTGATTACG | This paper | | |
| Sequence-based reagent | Primer for Q-RT-PCR Wdfy3_Mouse_R1: GTGGCCGTCTCCTTCAGTG | This paper | | |
| Sequence-based reagent | Primer for ChIP-Q-PCR Bchs_ChIP_F1: AGAACAGCTGTCTCGCACAA | This paper | | |
| Sequence-based reagent | Primer for ChIP-Q-PCR Bchs_ChIP_R1: CCTACATAGCGAGCAAGCGA | This paper | | |
| Sequence-based reagent | Primer for ChIP-Q-PCR DOR_ChIP_F1: ATTCGCTCGTCAGTCGTTGT | This paper | | |
| Sequence-based reagent | Primer for ChIP-Q-PCR DOR_ChIP_R1: TAACTGACGGGGGTGAGAGT | This paper | | |

*Continued on next page*

*Appendix 1—key resources table continued*

| Reagent type (species) or resource | Designation | Source or reference | Identifiers | Additional information |
|---|---|---|---|---|
| Sequence-based reagent | Primer for ChIP-Q-PCR Stat92E_ChIP_F1: AAGCGATCCA CATGCGATACT | This paper | | |
| Sequence-based reagent | Primer for ChIP-Q-PCR Stat92E_ChIP_R1: TCCTATCTTCC CGGTTTGGC | This paper | | |
| Recombinant DNA reagent | Plasmid: pUAST attb H3 | This paper | | |
| Software, algorithm | Microsoft Excel | Microsoft | https://www.microsoft.com/en-gb/ | |
| Software, algorithm | GraphPad Prism | GraphPad | https://www.graphpad.com/scientific-software/prism/ | |
| Software, algorithm | Adobe Illustrator | Adobe | https://www.adobe.com/uk/products/illustrator.html | |
| Software, algorithm | ImageJ | ImageJ | https://imagej.nih.gov/ij/ | |
| Software, algorithm | Leica LAS X-3D | Leica | https://www.leica-microsystems.com/de/produkte/mikroskop-software/details/product/leica-las-x-ls/ | |
| Software, algorithm | Imaris | Bitplane | http://www.bitplane.com/imaris | |
| Software, algorithm | Flexbar | *Dodt et al., 2012* | | |
| Software, algorithm | HiSat | *Kim et al., 2015* | | |
| Software, algorithm | SAMtools | *Li et al., 2009* | http://samtools.sourceforge.net/ | |
| Software, algorithm | StringTie | *Pertea et al., 2015* | | |
| Software, algorithm | R statistics package | R Core Team | https://www.r-project.org/ | |
| Other | Rapamycin gut RNA-seq analysed data | This paper | GEO: GSE148002 | |

