## [Decision Letter]

**Acceptance summary:**

This manuscript by Lu et al. demonstrates the role of histone protein H3 and H4 in aging using *Drosophila* as a model. They found rapamycin treatment could increase H3 and H4 protein expression which altered chromatin organization and induced autophagy in *Drosophila* intestine through transcriptional autophagy related genes regulation, leading to extended lifespan. Notably, these findings were conserved in mice as well. This work provides new insight and a new mechanism by which rapamycin functions to extend the lifespan and intestinal function.

**Decision letter after peer review:**

Thank you for submitting your article "A TORC1-histone axis regulates chromatin organisation and non-canonical induction of autophagy to ameliorate ageing" for consideration by *eLife*. Your article has been reviewed by 3 peer reviewers, and the evaluation has been overseen by a Reviewing Editor and Jessica Tyler as the Senior Editor. The reviewers have opted to remain anonymous.

The reviewers have discussed the reviews with one another and the Reviewing Editor has drafted this decision to help you prepare a revised submission.

Summary:

This manuscript by Lu et al. demonstrates the role of histone protein H3 and H4 in aging using *Drosophila* as a model. They found rapamycin treatment could increase H3 and H4 protein expression which altered chromatin organization and induced autophagy in *Drosophila* intestine through transcriptional autophagy related genes regulation, leading to extended lifespan. These findings were conserved in mice. All reviewers agree that this work is interesting and well-presented. However, reviewers have identified several major concerns and they all agree that these issues can be addressed by the authors with some additional experiments and some clarifications in a revised manuscript.

Essential revisions:

1. All three reviewers have found some confusing results/conclusions regarding rapamycin treatment/mTORC1 signaling and histone expression levels and requested clarifications.

a. It is not clearly why rapamycin treatment did not cause histone protein H3/H4 expression level change in brain, muscle or fat. This should be addressed in more details.

b. The link between histone levels and TORC1 activity is not clarified and is confusing. On the one hand, Rapamycin treatment leads to a repression of the age-related increase in H3/4 RNA levels in the intestine, on the other hand protein levels increase significantly after Rapamycin treatment, suggesting a strong activation of histone translation (while Rapamycin is known to suppress translation). Another possibility could be suppression of histone turnover, but there is no evidence shown for this. The authors need to find an explanation for this paradox, which is not even clearly discussed. Data explaining the Rapamycin induced increase in H3/4 protein levels is needed.

c. A possible explanation for the discrepancy highlighted above is that the histone protein levels measured by western blot from whole guts don't reflect cell-intrinsic changes, but changes in cell composition. The authors clearly show that ISC proliferation is affected by Rapamycin and by H3/4 over-expression, and that means that the number of cells in the gut changes significantly. While it is unclear how this would impact the overall levels of H3/4 protein at the tissue level, it is likely that the impact would be major, as new nuclei are being formed and the differentiation process of ECs involves polyploidization. The authors need to test how other perturbations that influence ISC proliferation influence overall H3/4 levels in the gut.

d. Another problem resulting from the discrepancy between the effects of Rapamycin on H3/4 transcript and protein levels is that the authors show that the effects of Rapamycin on chromatin organization and on longevity can be blocked by H3/4 knockdown. This is not logical: if Rapamycin treatment reduces H3/4 transcript levels, then a knockdown of H3/4 should, if at all, exacerbate the longevity effect. A related confusion emerges from the fact that aging guts show higher H3/4 protein levels, yet Rapamycin treatment increases these levels, so logically Rapamycin should make the guts 'older' and thus shorten lifespan?

e. What I don't see is a test of the ability of rapamycin to extend the lifespan of flies overexpressing H3/H4. If they work though the same pathway, there will likely be no synergy; but if they work through independent mechanisms, there will be synergy and an additive effect on lifespan. This would be similar to the previous work by Partridge (Bjedov et al. 2010) that showed DR and rapamycin have some additive effects, indicating overlapping but not identical mechanisms are engaged.

f. the authors do not test if DR and rapamycin have similar effects on histone expression but this would seem to be an obvious and important question to address, since per the labs previous work DR and rapamycin only work by overlapping, not identical mechanisms.

2. All reviewers have identified that some experiments and conclusions require further clarification.

a. In Figure 2A and B, rapamycin treatment seems to cause the increase of ECs number. What could be the reason?

b. In Figure 4, the authors showed the increase of histone modification H3K4me3 and H3K9me3 using Chip-qPCR after rapamycin treatment. What about related histone modification, such as H3K4me1 and H3K4me2, as well as the histone protein H4 modification?

c. The effect of Rapamycin on chromatin organization is interesting, but the data on nucleosome occupancy is not convincing. In Figure S3B, the signal at 1 min on MNase digest is similar between all conditions. It is unclear why the longer digest should provide a distinct signal. Maybe the authors need to explain this assay better and discuss how to best interpret this.

d. In Figure 2A, there is no age-related change in chromatin organization, even though H3/4 supposedly increase strongly in aging wt flies. This is contradictory.

e. While I see that overexpression of H3/H4 extends the lifespan of flies, this was not emphasized sufficiently in the text – I had to go looking for it – I see it mentioned at the end of the introduction, but not clearly stated in the abstract or discussion, or even I feel results. The failure of rapamycin to extend the lifespan of KD flies, which could be "sick", is far less important than the ability of H3/H4 overexpression to extend lifespan.

3. Two reviewers have identified experiments/analysis that require more robust/thorough analysis.

a. In Figure 5, more autophagy related genes, like atg1, should be tested, to strengthen the conclusion.

b. The authors use their RNAseq analysis simply to select a few autophagy related genes to study that are differentially expressed. This is arbitrary and ignores other transcriptional changes (the authors point out that there is no GO enrichment in their analysis). The induction of the autophagy related genes they select is not even very strong (barely 1.5fold induction of bchs for example). It is unclear whether these transcriptional changes really matter for the regulation of autophagy beyond the well-established direct effects of Rapamycin on autophagy through the repression of TORC1-mediated phosphorylation of ATG1.

c. To establish the role of H3/4 modulation in the transcriptional output of the Rapamycin treatment, the authors need to perform an RNAseq analysis of the H3/4 over-expression intestines.

---

## [Author Response]

Essential revisions:1. All three reviewers have found some confusing results/conclusions regarding rapamycin treatment/mTORC1 signaling and histone expression levels and requested clarifications.a. It is not clearly why rapamycin treatment did not cause histone protein H3/H4 expression level change in brain, muscle or fat. This should be addressed in more details.

We agree with the reviewers that this is important to address. Thus, we have tested the basal histone protein H3/H4 expression level in all tissues. Interestingly, we found that the basal protein expression level of histones was substantially higher in brain, muscle and fat than in intestine (Figure 1—figure supplement 1D). One possible interpretation is that rapamycin does not further increase histone levels in these three tissues because their chromatin is already fully occupied by histones. We have also addressed this in the Discussion.

b. The link between histone levels and TORC1 activity is not clarified and is confusing. On the one hand, Rapamycin treatment leads to a repression of the age-related increase in H3/4 RNA levels in the intestine, on the other hand protein levels increase significantly after Rapamycin treatment, suggesting a strong activation of histone translation (while Rapamycin is known to suppress translation). Another possibility could be suppression of histone turnover, but there is no evidence shown for this. The authors need to find an explanation for this paradox, which is not even clearly discussed. Data explaining the Rapamycin induced increase in H3/4 protein levels is needed.

We agree with the reviewers that the link between mTORC1 and histone levels needs clarification. To address this, we have performed several additional experiments:

1. We tested how rapidly histone levels increased in response to rapamycin treatment. We found that rapamycin substantially increased expression of histone proteins only two days after the treatment, without affecting their transcript levels (Figure 1—figure supplement 2A-B). This confirms our previous findings that rapamycin regulates histone protein levels post-transcriptionally.

2. We next investigated if rapamycin increased histone protein levels by inhibiting global translation by feeding flies with cycloheximide (1mM). Cycloheximide abolished the rapamycin-mediated increase in histone protein levels (Figure 2A), suggesting that rapamycin increases histone protein levels by increasing their translation.

Previous studies have demonstrated exceptions to translation suppression upon mTOR attenuation (Nandagopal and Roux et al. 2015), and histones are part of these exceptions, with their translational efficiencies increased (Thoreen et al. 2012). In these exceptional cases, translation is increased via eIF3 (Lee et al. 2015 and 2016). Moreover, these studies showed histone transcripts bound to eIF3 components directly, suggesting histones can be regulated by eIF3 (Lee et al. 2015). Therefore, we tested if eIF3 is required for the rapamycin-induced increased histone protein expression. We found that knocking down eIF3d or eIF3g in ECs abolished the increased histone protein levels upon rapamycin treatment (Figure 2B-C). In addition, knocking down eIF4e in ECs alone increased histone protein level similarly to rapamycin (Figure 2D), in line with the previous study that inhibition of eIF4 ensures mRNA translation occurs through an eIF3-specialised pathway (Lee et al. 2016). These results suggest rapamycin mediates histone proteins levels through the activity of translational factors eIF3 and eIF4.

3. We also tested if histone protein levels can be regulated by protein turnover. We found perturbed autophagy or proteasome activity had no effect on rapamycin induced increased histone protein levels (Figure 2—figure supplement 2A-B), and have addressed this in the Discussion.

c. A possible explanation for the discrepancy highlighted above is that the histone protein levels measured by western blot from whole guts don't reflect cell-intrinsic changes, but changes in cell composition. The authors clearly show that ISC proliferation is affected by Rapamycin and by H3/4 over-expression, and that means that the number of cells in the gut changes significantly. While it is unclear how this would impact the overall levels of H3/4 protein at the tissue level, it is likely that the impact would be major, as new nuclei are being formed and the differentiation process of ECs involves polyploidization. The authors need to test how other perturbations that influence ISC proliferation influence overall H3/4 levels in the gut.

We thank the reviewers for the suggestion. We have tested whether rapamycin treatment influenced cell composition and/or EC polyploidisation. Our results revealed that rapamycin affected neither cell composition nor polyploidisation even though it significantly reduced ISC mitosis (Figure 2—figure supplement 1A-C), suggesting increased expression of core histones in response to rapamycin treatment was caused by neither cell composition change nor EC polyploidization. The drop in ISC proliferation may instead reflect increased health and persistence of ECs.

We also tested other genetic manipulations that have been shown to influence ISC proliferation (Korzelius et al. 2014; Lu et al. 2021). While we observed that perturbations in Escargot expression and Sestrin activity dramatically reduced the pH3+ cell numbers in the fly intestine compared to their controls, they did not influence H3 and H4 protein levels in the fly intestine (Author response image 1). Manipulating ISC proliferation in other ways hence did not influence overall H3/4 levels in the intestine.

**Author response image 1. respfig1:** Perturbations of ISC proliferation via Escargot or Sestrin activity have no effect on histone protein levels in the fly intestine. (A-B) Adult-onset, ISC-specific knock-down of escargot by RNAi reduced ISC proliferation in intestine of flies (n = 15 intestines, Students t test, **p<0.01), but it had not effect on the expression of H3 and H4 at 20 days of age. (n = 4 biological replicates of 10 intestines per replicate, Students t test, ns p>0.05). The amount of proteins was normalized to DNA, shown by stain-free blot. (C-D) Sestrin mutant flies (SesnR407A) showed lower ISC proliferation in the intestine compared to wild type flies (n = 15 intestines, Students t test, ***p<0.001), but it had not effect on the expression of H3 and H4 at 20 days of age. (n = 4 biological replicates of 10 intestines per replicate, Students t test, ns p>0.05). The amount of proteins was normalized to DNA, shown by stain-free blot.

d. Another problem resulting from the discrepancy between the effects of Rapamycin on H3/4 transcript and protein levels is that the authors show that the effects of Rapamycin on chromatin organization and on longevity can be blocked by H3/4 knockdown. This is not logical: if Rapamycin treatment reduces H3/4 transcript levels, then a knockdown of H3/4 should, if at all, exacerbate the longevity effect. A related confusion emerges from the fact that aging guts show higher H3/4 protein levels, yet Rapamycin treatment increases these levels, so logically Rapamycin should make the guts 'older' and thus shorten lifespan?

We appreciate that this has not been sufficiently well-explained. As discussed in point 1b above, rapamycin treatment had no effect on histone transcript levels, while age did (Figure 1—figure supplement 2B-C). Age and rapamycin treatment mediated histone protein levels through independent mechanisms (Figure 1C, interaction p>0.05); importantly, rapamycin increased histone protein levels through post-transcriptional mechanisms. Rapamycin treatment attenuated the increase in histone transcript levels at older ages, possibly because it attenuated ageing per se. Directly knocking down H3 or H4 transcription in adult ECs blocked the rapamycin-dependent increase in histone protein levels and consequently blocked the beneficial effects of rapamycin on gut health and longevity (Figure 4A-B, D-E, G).

To address the last point, the increase in histone protein level with age was subtle and much lower than that induced by rapamycin treatment; where at all ages, histone protein levels were strongly elevated compared to controls (Figure 1C). The subtle increase in histone proteins observed over ageing did not result in a marked chromatin reorganisation (Figure 3A). We have clarified this point in the Discussion.

e. What I don't see is a test of the ability of rapamycin to extend the lifespan of flies overexpressing H3/H4. If they work though the same pathway, there will likely be no synergy; but if they work through independent mechanisms, there will be synergy and an additive effect on lifespan. This would be similar to the previous work by Partridge (Bjedov et al. 2010) that showed DR and rapamycin have some additive effects, indicating overlapping but not identical mechanisms are engaged.

We already showed that the combination of rapamycin and overexpression of H3/H4 did not further extend lifespan or improve gut health (Figure 4C, F, H), suggesting they work through the same pathway. Now, we have better emphasised this result in the manuscript to clarify this point.

f. the authors do not test if DR and rapamycin have similar effects on histone expression but this would seem to be an obvious and important question to address, since per the labs previous work DR and rapamycin only work by overlapping, not identical mechanisms.

We thank the reviewers for this suggestion. We have further tested whether DR affected histone expression by comparing the histone H3/4 protein expression levels upon 1xSYA and 2xSYA food. We found that histone H3/4 protein levels were not affected by the food concentration (Figure 1—figure supplement 3).

2. All reviewers have identified that some experiments and conclusions require further clarification.a. In Figure 2A and B, rapamycin treatment seems to cause the increase of ECs number. What could be the reason?

ECs were significantly smaller in the rapamycin fed females (Author response image 2), so there are more ECs per unit gut volume.

**Author response image 2. respfig2:** Rapamycin reduces cell size of enterocytes. (A) Cell size of enterocytes in *w^Dah^* females significantly reduced upon rapamycin treatment at 28 days of age. (n = 10 intestines, n = 10-20 ECs were observed per intestine, Student t-test, **p<0.01).

b. In Figure 4, the authors showed the increase of histone modification H3K4me3 and H3K9me3 using Chip-qPCR after rapamycin treatment. What about related histone modification, such as H3K4me1 and H3K4me2, as well as the histone protein H4 modification?

We thank the reviewers for this suggestion. We have added data on another histone modification, H3K27me3, to strengthen our findings (Figure 5B). Since the scope of this paper is to show increased histone proteins regulated Bchs transcript through histone modifications, but not to investigate which specific histone modifications play the key role in this process, we consider our data is sufficient.

c. The effect of Rapamycin on chromatin organization is interesting, but the data on nucleosome occupancy is not convincing. In Figure S3B, the signal at 1 min on MNase digest is similar between all conditions. It is unclear why the longer digest should provide a distinct signal. Maybe the authors need to explain this assay better and discuss how to best interpret this.

Thank you for highlighting this. We have shown the signal of mono, di and tri-nucleosomes at 1min on MNase digest from rapamycin and/or RU treatments was significantly stronger than the control, and extending the MNase digest time led to more di- and tri-nucleosomes becoming mono- nucleosomes (Figure 3—figure supplement 1B). We have clarified our explanation in the manuscript.

d. In Figure 2A, there is no age-related change in chromatin organization, even though H3/4 supposedly increase strongly in aging wt flies. This is contradictory.

Thank you for highlighting that our discussion of this was not sufficiently clear. We have shown that histone proteins protein levels increased over ageing, but to a much lower extent than upon rapamycin treatment, at all ages (Figure 1C). Thus, the ageing change may not be sufficient to cause a marked chromatin reorganisation. We have included this in the Discussion.

e. While I see that overexpression of H3/H4 extends the lifespan of flies, this was not emphasized sufficiently in the text – I had to go looking for it – I see it mentioned at the end of the introduction, but not clearly stated in the abstract or discussion, or even I feel results. The failure of rapamycin to extend the lifespan of KD flies, which could be "sick", is far less important than the ability of H3/H4 overexpression to extend lifespan.

We agree with the reviewers that data on overexpression of H3/H4 is insufficiently emphasised, and have corrected this in our manuscript.

3. Two reviewers have identified experiments/analysis that require more robust/thorough analysis.a. In Figure 5, more autophagy related genes, like atg1, should be tested, to strengthen the conclusion.

We agree that further manipulations of autophagy-related genes bear testing to strengthen the conclusion. We have tested Atg1 RNAi and found it caused a similar phenotype to Atg5 RNAi. We have included the result in our manuscript (Figure 6—figure supplement 1A-B).

b. The authors use their RNAseq analysis simply to select a few autophagy related genes to study that are differentially expressed. This is arbitrary and ignores other transcriptional changes (the authors point out that there is no GO enrichment in their analysis). The induction of the autophagy related genes they select is not even very strong (barely 1.5fold induction of bchs for example). It is unclear whether these transcriptional changes really matter for the regulation of autophagy beyond the well-established direct effects of Rapamycin on autophagy through the repression of TORC1-mediated phosphorylation of ATG1.

We thank the reviewers for the suggestion. In the manuscript, we did not detect any significant enrichment of specific biological processes by GO enrichment analysis, we therefore selected autophagy related genes based on intensive previous studies that showed autophagy was playing a key role in the gut health and longevity (Hansen et al. 2018). We did not rule out that other transcriptional changes would contribute to the H3/4 effects on longevity, but we feel it would be better to explore these possibilities in follow-up studies.

In addition, we agree with the reviewers’ comment that we had not clarified the link between transcriptional changes for the regulation of autophagy and the TORC1-mediated phosphorylation of Atg1. Thus, we have tested if knock-down or overexpression of Bchs influenced the phosphorylation status of Atg1. We found that neither of these manipulations affected the phosphorylation of Atg1(Figure 7—figure supplement 2A-B), in line with a previous study that suggested Bchs to be acting downstream of Atg1 (Sim et al. 2019). These results suggest that Bchs was sufficient to mediate autophagy without affecting mTORC1-mediated phosphorylation of Atg1.

We have included these points in our manuscript.

c. To establish the role of H3/4 modulation in the transcriptional output of the Rapamycin treatment, the authors need to perform an RNAseq analysis of the H3/4 over-expression intestines.

We do not believe RNAseq analysis of intestines over-expressing H3/4 would add significant value to our manuscript, as we would be unlikely to obtain biologically significant data from the comparison of two RNAseq datasets in which only a handful of transcriptional changes occur under rapamycin treatment. We have instead focused on reinforcing the mechanism underlying H3/4-induced autophagy, as described in point 3b, which includes epistasis analysis of H3/H4 and rapamycin (Figure 5, Figure 5—figure supplement 2 and 3).

References:

Lee, A.S., Kranzusch, P.J., and Cate, J.H. (2015). eIF3 targets cell-proliferation messenger RNAs for translational activation or repression. Nature 522, 111-114.

Lee, A.S., Kranzusch, P.J., Doudna, J.A., and Cate, J.H. (2016). eIF3d is an mRNA cap-binding protein that is required for specialized translation initiation. Nature 536, 96-99.

Nandagopal, N., and Roux, P.P. (2015). Regulation of global and specific mRNA translation by the mTOR signaling pathway. Translation (Austin) 3, e983402.

Thoreen, C.C., Chantranupong, L., Keys, H.R., Wang, T., Gray, N.S., and Sabatini, D.M. (2012). A unifying model for mTORC1-mediated regulation of mRNA translation. Nature 485, 109-113.

Korzelius, J., Naumann, S.K., Loza-Coll, M.A., Chan, J.S., Dutta, D., Oberheim, J., Glasser, C., Southall, T.D., Brand, A.H., Jones, D.L., et al. (2014). Escargot maintains stemness and suppresses differentiation in Drosophila intestinal stem cells. The EMBO journal 33, 2967-2982.

Lu, J., Temp, U., Müller-Hartmann, A. Esser, J., Grönke, S., Partridge, L., (2021) Sestrin is a key regulator of stem cell function and lifespan in response to dietary amino acids. Nat Aging 1, 60–72.

Hansen, M., Rubinsztein, D.C., and Walker, D.W. (2018). Autophagy as a promoter of longevity: insights from model organisms. Nature Reviews Molecular cell biology. 19, 579-593.

Sim, J., Osborne, K.A., Argudo Garcia, I., Matysik, A.S., and Kraut, R. (2019). The BEACH Domain Is Critical for Blue Cheese Function in a Spatial and Epistatic Autophagy Hierarchy. Front Cell Dev Biol 7, 129.